# NOVA-dependent regulation of cryptic NMD exons controls synaptic protein levels after seizure

**Taesun Eom[1], Chaolin Zhang[1], Huidong Wang[1], Kenneth Lay[1], John Fak[1], Jeffrey L Noebels[2], Robert B Darnell[1,3]\***

[1]Laboratory of Molecular Neuro-Oncology, Rockefeller University, New York, United States; [2]Developmental Neurogenetics Laboratory, Department of Neurology, Baylor College of Medicine, Houston, United States; [3]Howard Hughes Medical Institute, Rockefeller University, New York, United States

**Abstract** The neuronal RNA binding protein NOVA regulates splicing, shuttles to the cytoplasm, and co-localizes with target transcripts in dendrites, suggesting links between splicing and local translation. Here we identified >200 transcripts showing NOVA-dependent changes in abundance, but, surprisingly, HITS-CLIP revealed NOVA binds these RNAs in introns rather than 3′ UTRs. This led us to discover NOVA-regulated splicing of cryptic exons within these introns. These exons triggered nonsense mediated decay (NMD), as UPF1 and protein synthesis were required for NOVA's effect on RNA levels. Their regulation was dynamic and physiologically relevant. The NMD exons were regulated by seizures, which also induced changes in Nova subcellular localization and mediated large changes in synaptic proteins, including proteins implicated in familial epilepsy. Moreover, Nova haploinsufficient mice had spontaneous epilepsy. The data reveal a hidden means of dynamic RNA regulation linking electrical activity to splicing and protein output, and of mediating homeostatic excitation/inhibition balance in neurons.

**\*For correspondence:** darnelr@rockefeller.edu

**Reviewing editor**: Benjamin Blencowe, University of Toronto, Canada

## Introduction

RNA binding proteins (RNABPs) play critical roles in the brain to regulate neuronal development and activity, as underscored by the finding that dysregulation of RNABP function underlies a growing list of neurological disorders (*Licatalosi and Darnell, 2006*; *Cooper et al., 2009*; *Poulos et al., 2011*). This connection was first highlighted by the use of autoantibodies present in the paraneoplastic neurologic disorders (*Darnell, 1996*, *2004*; *Darnell and Posner, 2006*, *2011*) to discover that NOVA and ELAVL/Hu are neuron-specific RNABPs (*Szabo et al., 1991*; *Buckanovich et al., 1993*; *Okano and Darnell, 1997*; *Kim et al., 2009*; *Dredge and Jensen, 2011*). The characterization of these proteins, as well as a number of subsequently described mammalian RNABPs, including NeuN/RBFOX3 (*Kim et al., 2009*; *Dredge and Jensen, 2011*) and brPTB/nPTB/PTBP2 (*Markovtsov et al., 2000*; *Polydorides et al., 2000*), suggests that the brain, and neurons in particular, harbor unique sets of RNABPs to uniquely regulate RNA metabolism (*Darnell, 2006*).

NOVA proteins are among the best-studied mammalian tissue-specific RNABPs. Two paralogs, NOVA1 (*Buckanovich et al., 1993*) and NOVA2 (*Yang et al., 1998*), are present in mammals, with largely indistinguishable biochemical properties but largely reciprocal expression within the central nervous system (*Darnell, 2006*). They bind to YCAY-rich elements (*Buckanovich and Darnell, 1997*; *Jensen et al., 2000a*; *Lewis et al., 2000*; *Ule et al., 2006*) to regulate neuronal alternative splicing (*Polydorides et al., 2000*; *Jensen et al., 2000b*; *Dredge and Darnell, 2003*; *Dredge et al., 2005*; *Ule et al., 2005b*; *Licatalosi et al., 2008*) and mRNA transport of some target RNAs (*Racca et al., 2010*).

**eLife digest** After the DNA in a gene has been transcribed into messenger RNA, portions of the mRNA called introns are removed, and the remaining stretches of mRNA, which are known as exons, are spliced together. Within eukaryotic cells, a process known as alternative splicing allows a single gene to encode for multiple protein variants by ensuring that some exons are included in the final, modified mRNA, while other exons are excluded. This modified mRNA is then translated into proteins.

Eukaryotic cells also contain proteins that bind to RNA to regulate alternative splicing. These RNA-binding proteins are often found in both the cytoplasm and nucleus of cells, and their involvement in splicing may be linked to other processes in the cell such as mRNA localization and translation. It has also become clear over the past two decades that certain types of RNA-binding proteins, including NOVA proteins, are only found in neurons, and that these proteins have been best characterized as alternative splicing regulators. Recent work has also suggested that they also have important roles in regulating neuronal activity and development, and that their actions in neuronal nuclei and cytoplasm might be coordinated.

Now Eom et al. use the predictive power of a high throughput sequencing and crosslinking method termed HITS-CLIP to show that NOVA proteins can indirectly regulate cytoplasmic mRNA levels by regulating the process of alternative splicing in the nucleus to produce 'cryptic' exons in the brains of mice. The presence of these exons in the mRNA leads to the production of premature termination codons in the cytoplasm. These codons trigger a process called nonsense-mediated decay that involves identifying mRNA transcripts that contain nonsense mutations, and then degrading them. These cryptic exons were seen in mice missing the NOVA proteins, where they are expressed in abnormally high levels; in normal mice, these exons have not been seen before, hence they were termed 'cryptic'.

Eom et al. also show that these cryptic exons are physiologically relevant by inducing epileptic seizures in mice. Following the seizures, they find that the NOVA proteins up-regulate and down-regulate the levels of different cryptic exons, leading to changes in the levels of the proteins encoded by these mRNAs, including proteins that inhibit further seizures. Overall the results indicate that, by controlling the production of various proteins in neurons, these previously unknown cryptic exons have important roles in the workings of the brain.

NOVA-mediated RNA regulation is essential, as *Nova* DKO (*Nova1*−/− and *Nova2*−/−) mice die immediately at birth (***Ruggiu et al., 2009***), although the range of NOVA actions is still being enumerated (***Zhang et al., 2010***). For example, NOVA, like many other splicing factors (***Huang and Steitz, 2005***), shuttles between the nucleus and cytoplasm (***Racca et al., 2010***), reflecting the interesting but incompletely understood relationship between the regulation of RNA splicing, transport, stability, localization and translation (***Maniatis and Reed, 2002***; ***Huang and Steitz, 2005***; ***Martin and Ephrussi, 2009***; ***Darnell, 2010***).

Several well-defined RNA elements mediate RNA stability and localization, such as 3′ UTR adenine/uridine-rich (ARE) elements recognized by the Elavl family of RNABPs (***Levine et al., 1993***; ***Myer et al., 1997***; ***Peng et al., 1998***; ***Brennan and Steitz, 2001***; ***Kishore et al., 2010***) and the zipcode recognized by ZBP1 (***Chao et al., 2010***). NOVA may play coordinate roles in the nucleus and cytoplasm, as HITS-CLIP studies have shown direct binding to introns and 3′ UTRs of *GlyRa2* and *Girk2* mRNA (***Racca et al., 2010***), and evidence of splicing and a role for localization in each. However, the extent and means by which NOVA might mediate actions in both compartments remains uncertain.

Here we explore the relationship between nuclear and cytoplasmic functions of NOVA by undertaking HITS-CLIP (***Darnell, 2010***) on each fraction separately, and comparing results with microarray analysis of RNA in *Nova*-null mice. NOVA crosslinks to many transcripts in both nuclear/intronic and cytoplasmic/3′ UTR clusters, suggesting an ordered set of cis-actions on target RNAs. Surprisingly, however, among 229 transcripts showing significant steady-state changes, NOVA binding was primarily present in intronic rather than cytoplasmic elements. This led us to hypothesize and identify previously unannotated exons or alternative splicing patterns near these intronic elements that introduced premature termination codons (PTC) able to mediate nonsense-mediated mRNA decay (NMD). Thus HITS-CLIP reveals that NOVA regulates otherwise cryptic NMD exons

in the nucleus to indirectly regulate steady-state mRNA levels in the cytoplasm. We put this finding in a physiologic context, showing that these exons encode synaptic proteins, they are regulated by seizure activity, and that Nova itself is important in epilepsy, as the protein shifts from the neuronal nucleus to cytoplasm shortly after seizure, and Nova haploinsufficient mice have spontaneous epilepsy. Together, these data suggest a role for Nova in maintaining homeostasis after seizure, in part through the regulation of cryptic NMD exons.

## Results

### Changes in steady-state transcript levels in NOVA DKO brain

Previous CLIP experiments have identified NOVA binding sites near alternative exons and poly(A) sites (*Ule et al., 2003*; *Ule et al., 2005b*, *2006*; *Licatalosi et al., 2008*), but have not fully addressed whether NOVA might also regulate the steady-state level of some transcripts. To assess the degree to which NOVA might regulate this aspect of neuronal mRNA metabolism, we compared steady-state levels of total poly(A)+ mRNAs in four *Nova* WT and four *Nova1*$^{-/-}$/*2*$^{-/-}$ (double KO; DKO) brains from P0 littermates (DKO mice die shortly after birth) with Affymetrix exon arrays. A total of 229 genes showing significant changes ($|\log_2(WT/DKO)|>0.3$; $p < 0.05$, Student's t-test) were identified (*Figure 1A*). Interestingly, this data indicated that NOVA had a disproportionate action to increase (rather than decrease) steady state levels of brain transcripts. Specifically, 215 of the 229 transcripts showed significant decreases in the absence of NOVA (NOVA-dependent 'upregulation' of steady-state mRNA levels; *Figure 1A*). 22 showed a >1.5-fold change, and 11 of these were validated experimentally by qRT-PCR (*Figure 1B*; *Table 1*). Only 14 transcripts showed comparable NOVA-dependent mRNA suppression of steady-state levels (five showing a greater than 1.5-fold change), and these were validated experimentally (*Figure 1C* and *Table 1*).

These observations do not indicate whether NOVA acts directly or indirectly on these 229 transcripts to regulate their levels. HITS-CLIP has proven to be a reliable means of defining sites of direct functional RNA–protein interactions (*Licatalosi and Darnell, 2010*). Analysis of prior CLIP data (*Licatalosi et al., 2008*; *Zhang et al., 2010*) demonstrated that many of the 229 transcripts showing NOVA-dependent changes in level had CLIP tags when whole brain extracts were crosslinked (see also below). Such changes could result from nuclear and/or cytoplasmic actions, and we have previously found evidence that NOVA may act in both compartments (*Racca et al., 2010*). Therefore, we undertook a systematic comparison of NOVA binding in each compartment.

### Nuclear and cytoplasmic NOVA HITS-CLIP

To define NOVA-RNA interactions in the nucleus and cytoplasm separately, we crosslinked mouse brain cortex, 'freezing' RNA–protein complexes for analysis, and then subfractionated tissue into nuclear and cytoplasmic fractions. To assess the integrity of this fractionation, we performed immunoblot analysis with markers known to be specific for each fraction and antibodies to NOVA, demonstrating that each fraction was >90% pure, and that ~60% of NOVA was nuclear and 40% cytoplasmic (*Figure 1D*, consistent with and detailed in a previous study (*Racca et al., 2010*)). While this fractionation scheme (based on salt extraction) gave a relatively clean separation between each compartment, we cannot exclude the possibility that some NOVA protein was not extracted from each compartment under these conditions. Nonetheless, these data suggested that highly enriched NOVA-RNA complexes present in subcellular compartments could be purified and analyzed.

We examined NOVA RNA binding maps to nuclear and cytoplasmic RNAs by analyzing HITS-CLIP results from each fraction (*Figure 1E* and *Figure 1—figure supplement 1*; see also *Table 2*). We defined NOVA CLIP tag clusters by grouping overlapping RNA tags identified in multiple biologic replicates (biologic complexity (BC) > 2) (*Licatalosi et al., 2008*) from four independent animals. These clusters were classified according to their locations within protein-coding transcripts, using annotations from cDNA/EST data and RefSeq and UCSC known genes (*Zhang et al., 2010*). By focusing on CLIP tags in robust clusters (BC = 4, peak height (PH) ≥ 4), we observed that nuclear HITS-CLIP tags were enriched in introns (90%) and cytoplasmic HITS-CLIP tags in 3′ UTR (63%, including ~13% within 10 kb downstream of annotated transcripts; *Figure 1F*; see (*Licatalosi et al., 2008*)). NOVA cytoplasmic CLIP data also contained a significant number of intronic clusters, but many of these were derived from abundant non-coding RNA species, such as ribosomal RNAs (also evident in nuclear CLIP data). After

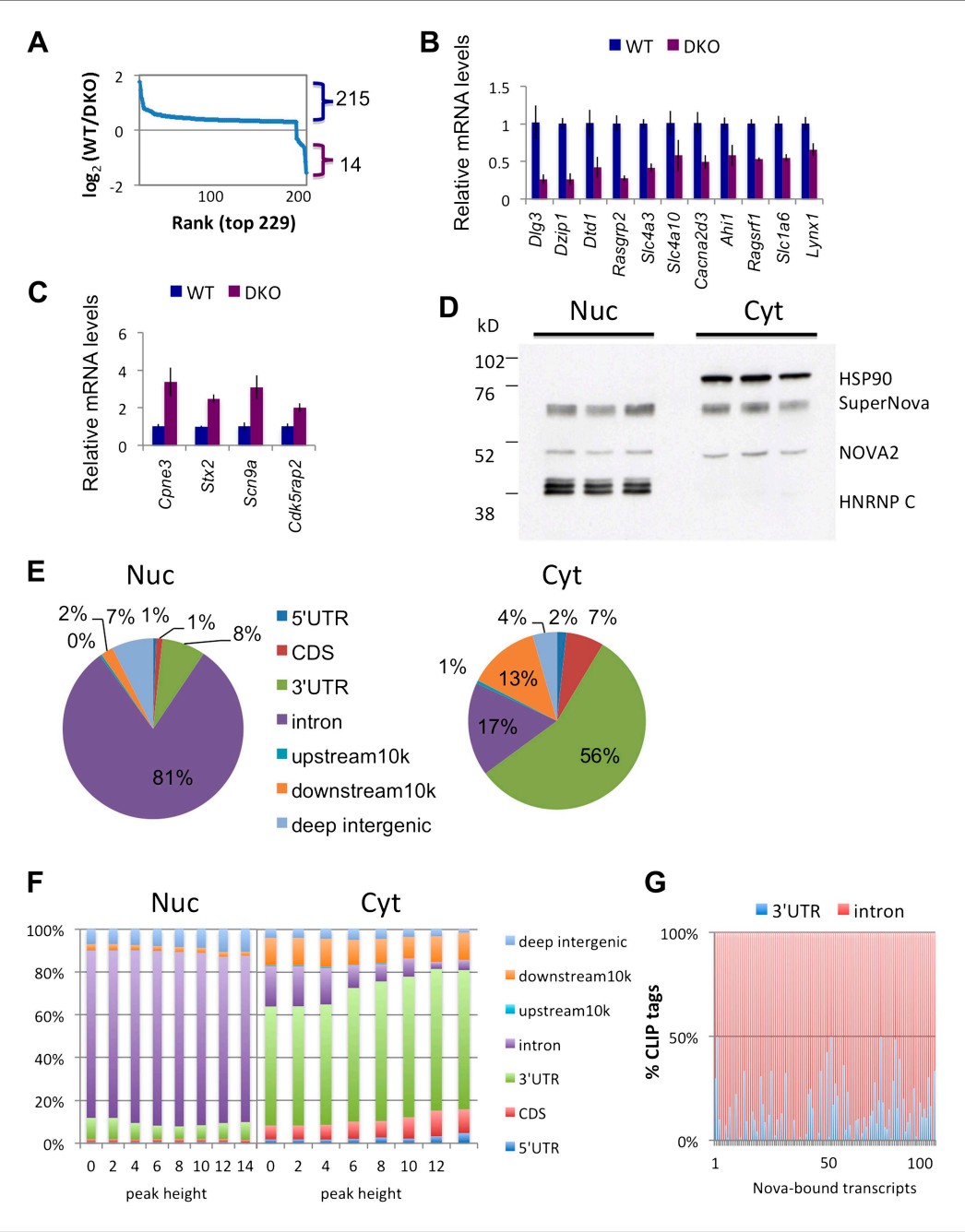

**Figure 1**. NOVA proteins up/down-regulate transcript levels. (**A**) Affymetrix exon arrays were interrogated with RNA from WT vs DKO E18.5 mouse whole brains, and normalized transcript intensities were plotted in $\log_2$ scale ($\log_2$(WT/DKO) > 0.3 or −0.3 and p<0.05). The X-axis indicates ranks of transcripts from the top and Y-axis is the measure of relative transcript levels (WT/DKO) in $\log_2$ scale. Blue bracket represents transcripts whose levels are increased in WT relative to DKO brain (NOVA-dependent 'upregulation' of steady-state mRNA levels), and the purple bracket represents down-regulated transcripts in WT relative to DKO (NOVA-dependent repression of steady-state mRNA levels). (**B**),(**C**) qRT-PCR data of representative NOVA regulated transcripts. Y-axis represents the mRNA levels in which WT is normalized to 1.0. Data is from three biologic replicates (three animals) and three technical replicates (nine reactions per point); error bars represent standard deviation. For each point p<0.001; see *Table 1* for additional data. NOVA up-regulated transcripts in WT versus DKO (**B**; corresponding to blue bracket in Figure 1A) and NOVA down-regulated transcripts (**C**; corresponding to purple bracket in Figure 1A) are shown. (**D**) Immunoblot analysis of NOVA distribution in nuclear and cytoplasmic fractions from mouse brain irradiated by UV. Each lane represents the different brain extracts as biological replicates. HSP90 is used as a cytoplasmic

*Figure 1. Continued on next page*

*Figure 1. Continued*

marker, and hnRNP-C1/C2 as a nuclear marker. The NOVA2 antibody detects both large and small NOVA2 isoforms (***Yang et al., 1998***). (**E**) Breakdown of BC = 4 clusters for nuclear and cytoplasmic Nova HITS-CLIP. Downstream 10K clusters are enriched in unannotated 3′ UTRs (***Licatalosi et al., 2008***); see also ***Table 2***. (**F**) Distribution of BC = 4 clusters by peak height for both nuclear and cytoplasmic HITS-CLIP; more stringent cytoplasmic clusters show enrichment in 3′ UTR. (**G**) Distribution of CLIP tags (intronic, red; 3′ UTR, blue) from the list of NOVA up-regulated RNAs. Each point on the X-axis represents a Nova-dependent gene (in arbitrary order) and Y-axis represents the percentage of intronic/3′ UTR tags for transcripts with total tags >5.
The following figure supplements are available for figure 1:

**Figure supplement 1**. Nova CLIP results.

**Figure supplement 2**. Correlations: analysis of Nova CLIP results.

we reanalyzed the clusters with repetitive sequences excluded, only 17% clusters were located in introns (***Figure 1F***). In addition, when we analyzed the distribution of clusters as a function of peak height (PH), we identified a large proportion of intronic clusters in nuclear CLIP data, while less than 5% of NOVA cytoplasmic CLIP tag clusters were intronic (at PH 12–14).

The proportion of apparent cytoplasmic NOVA binding sites in introns was nonetheless an unexpectedly high number, prompting us to repeat the cytoplasmic purification with an independent method (based upon differential centrifugation; data not shown) and then repeat HITS-CLIP analysis on mouse brain. This experiment independently confirmed the presence and position of NOVA2 cytoplasmic ($R^2 = 0.77$) and nuclear ($R^2 = 0.77$) CLIP tags between experiments. We have not studied the function of the small numbers of reproducible NOVA 'cytoplasmic' intronic clusters further, which most likely reflect remaining impurity of the cell fractionation, but could also reflect other processes, including NOVA binding to non-coding RNAs, stable processed introns, or unspliced transcripts. Taken together, our data suggest that NOVA binds predominantly to intronic elements in pre-mRNA in the nucleus and mature mRNA elements in the cytoplasm, but may have an unexpected role in binding intronic non-coding RNA elements in the cytoplasm as well.

We previously identified RNA transcripts bound by NOVA in the nucleus that also co-localize with (*GlyRa2*) or are localized by (*Girk2*) NOVA in neuronal processes (***Racca et al., 2010***), prompting us to compare the NOVA nuclear and cytoplasmic binding maps. We first compared the frequency and extent to which cytoplasmic NOVA clusters were also present in the same location in nuclear CLIP. There was a significant albeit small correlation between cytoplasmic and nuclear HITS-CLIP tags ($R^2 = 0.32$, ***Figure 1—figure supplement 2***) and robust clusters (BC = 4; $R^2 = 0.36$, ***Figure 1—figure supplement 2***). Moreover, when we relaxed the stringency of the nuclear clusters, we found that in most transcripts with robust cytoplasmic clusters, NOVA was already bound to those clusters in the nucleus. Among 504 transcripts with robust cytoplasmic clusters (BC $\geq$ 4, PH > 10), 235 had robust nuclear clusters (BC $\geq$ 4, PH > 10), and an additional 251 (46.6%) had weaker nuclear clusters. There were 18 unique cytoplasmic clusters (i.e. clusters not present in nuclear CLIP, but in transcripts which nonetheless had nuclear clusters in other locations), suggesting the possibility of a reassortment of NOVA binding sites in cis on individual transcripts; such a mechanism might allow nuclear and cytoplasmic functions like RNA localization or RNA stability to be linked (see ***Racca et al., 2010***).

We analyzed the relationship between NOVA cytoplasmic and nuclear binding in transcripts whose levels are NOVA-dependent. One common mechanism by which transcript steady-state levels are regulated is through actions on stability elements within the 3′ UTR, either through the actions of miRNAs (***Fabian et al., 2010***) or through RNABP-RNA regulatory interactions via actions on conserved elements such as ARE sequences (***Brennan and Steitz, 2001***; ***Yang et al., 2003***; ***Khabar, 2010***). Surprisingly however, the distribution of NOVA CLIP tags within NOVA-regulated transcripts (specifically, in the 149 of 229 transcripts harboring $\geq$5 CLIP tags) revealed that the majority of NOVA CLIP tags (~76%) were not present on the 3′ UTRs of NOVA-regulated transcripts. Instead the predominant location of NOVA CLIP tags on these transcripts was intronic (***Figure 1G***), suggesting the possibility that NOVA action on pre-mRNA might be linked to steady-state levels of NOVA target mRNAs.

**Table 1.** Validation data for transcripts showing Nova-dependent changes in steady-state mRNA levels

| qRT-PCR fold | Microarray-predicted fold | p-value | Gene |
|---|---|---|---|
| Transcripts with larger predicted fold changes | | | |
| 3.85 ± 0.84 | 3.39 | <0.001 | **Dlg3** (Sap102) |
| 3.79 ± 0.29 | 2.73 | <0.001 | **Dzip1** |
| 2.40 ± 0.43 | 2.25 | <0.001 | Dtd1 |
| 3.64 ± 0.39 | 2.23 | <0.001 | Rasgrp2 |
| 2.42 ± 0.15 | 1.97 | <0.001 | **Slc4a3** |
| 1.74 ± 0.28 | 1.81 | <0.001 | **Slc4a10** |
| 2.06 ± 0.3 | 1.72 | <0.001 | Cacna2d3 |
| 1.73 ± 0.14 | 1.71 | <0.001 | **Ahi1** |
| 1.90 ± 0.12 | 1.68 | <0.001 | **Rasgrf1** |
| 1.83 ± 0.18 | 1.67 | <0.001 | Slc1a6 |
| 1.53 ± 0.13 | 1.67 | <0.001 | Lynx1 |
| 0.50 ± 0.07 | 0.68 | <0.001 | **Cdk5rap2** |
| 0.40 ± 0.02 | 0.54 | <0.001 | **Stx2**(syntaxin 2) |
| 0.30 ± 0.04 | 0.45 | <0.001 | Cpne3(copine III) |
| 0.33 ± 0.07 | 0.34 | <0.001 | **Scn9a** |
| Transcripts with smaller predicted fold changes | | | |
| 1.98 ± 0.16 | 1.48 | <0.001 | Syt2 |
| 1.41 ± 0.1 | 1.47 | <0.001 | **Actl6b** |
| 1.74 ± 0.07 | 1.40 | <0.001 | Gria3 |
| 1.56 ± 0.09 | 1.30 | <0.001 | Syngr3* |
| 1.53 ± 0.13 | 1.29 | <0.001 | Glrb* |
| 1.33 ± 0.20 | 1.25 | <0.001 | Gabbr1* |
| 0.69 ± 0.07 | 0.77 | <0.001 | **Plekha5** |

Summary of validation data for transcripts showing Nova-dependent change in steady-state mRNA levels. Upper table: validation data for transcripts showing larger fold predicted changes by microarray. Lower table: validation data for transcripts showing small predicted changes by microarray, to illustrate sensitivity of the data. In each table, the first column shows fold change from qRT-PCR, the second column shows fold change from Affymetrix-exon array data, the third column shows the p value and the fourth column shows the name of gene. Data for each gene is derived from three biologic replicates (three animals) and three technical replicates (nine reactions per point); these data are statistically significant (p<0.001; Student's t-test). Genes in bold are cryptic exons confirmed by sequencing (see *Figure 3—source data 1* and *Figure 4D*). Genes with asterisk are those with robust 3′ UTR clusters (see *Figure 8*).

### *Dlg3 (SAP102)* mRNA and protein levels are reduced in the absence of NOVA

To address the mechanism by which NOVA regulates mRNA steady-state levels, we analyzed individual targets in more detail. The transcript encoding *Dlg3* (*Sap102*) showed the most significant NOVA-dependent upregulation (i.e. the largest decrease in steady state levels in NOVA DKO brain) in exon arrays (*Figure 1B,C*), which was confirmed in qRT-PCR studies (*Table 1*). The *Dlg3* transcript had a large number of NOVA CLIP tags (*Figure 2A*), suggesting that it might be both directly bound and regulated by NOVA. Consistent with this possibility, we found a nearly 10-fold reduction in *Dlg3* mRNA in NOVA DKO brain RNA samples by Northern blot analysis using two different probes and semi-quantitative RT-PCR (*Figure 2B*, *Figure 2—figure supplement 1*), with intermediate changes seen in single NOVA1 or NOVA2 KO mice (data not shown).

To assess whether these changes in mRNA levels led to corresponding changes in DLG3 protein levels, we analyzed protein extracts from NOVA WT vs DKO brain. Western blots demonstrated that

**Table 2.** Summary of CLIP tag mapping data

| Sample | Tag total | Tag mappable | Tag unique |
|---|---|---|---|
| Nuclear 1 | 1,504,345 | 1,031,115 | 429,527 |
| Nuclear 2 | 4,984,997 | 3,161,194 | 363,771 |
| Nuclear 3 | 7,184,066 | 3,944,912 | 214,982 |
| Nuclear 4 | 6,656,628 | 3,144,744 | 325,507 |
| Nuclear sum | | | 1,333,787 |
| Cytoplasm 1 | 1,454,697 | 975,222 | 89,463 |
| Cytoplasm 2 | 4,751,331 | 2,780,173 | 37,027 |
| Cytoplasm 3 | 3,653,600 | 2,005,202 | 36,228 |
| Cytoplasm 4 | 3,813,960 | 2,382,946 | 39,145 |
| Cytoplasmic sum | | | 201,863 |

DLG3 protein levels were severely reduced (~90% reduction; *Figure 2C*) in DKO brain, closely reflecting the reduction in *Dlg3* mRNA. We also examined DLG3 protein by immunofluorescence microscopy using anti-DLG3 antibodies. We co-cultured WT/DKO neurons together and stained for NOVA protein to distinguish WT from DKO neurons, using Neurofilament-M (NF-M) immunoreactivity as a positive control to stain all neurons. These experiments clearly showed that NOVA DKO neurons had very little DLG3 signal (*Figure 2D*). Taken together, these data demonstrate that the absence of NOVA protein in neurons is associated with a drastic reduction in *Dlg3* mRNA levels and a concomitant reduction in DLG3 protein levels.

To address whether NOVA protein might directly regulate *Dlg3* mRNA levels, we examined the specific locations of NOVA HITS CLIP tags along the *Dlg3* transcript (*Figure 2A*). NOVA binding sites were present in both the 3′ UTR and introns, and in both locations these sites harbored NOVA binding elements (clusters of the sequence YCAY; (*Zhang et al., 2010*)). To assay the functional outcome of NOVA binding to the 3′ UTR elements, we generated a construct encoding a destabilized d1EGFP fused to either a 214 nt fragment (NOVA binding clusters in 3′ UTR shown in *Figure 2A*) from the wild-type *Dlg3* 3′ UTR or a mutant fragment which would disrupt NOVA binding (YCAY->YAAY; *Figure 3—figure supplement 1*). 24 hr after transfection of these constructs into Neuro2a (N2a) neuroblastoma cells, actinomycin D was added for 5 hr in order to block transcription, and *Dlg3* mRNA levels were followed by qRT-PCR (*Figure 3—figure supplement 2*). These data revealed a relatively small but clear decrease in d1EGFP mRNA containing mutant YAAY NOVA binding sites (~50% reduction) compared to transcripts harboring WT YCAY elements, indicating that NOVA binding to the 3′ UTR had a small effect on *Dlg3* steady state levels. However, this interaction did not appear to be sufficient to explain the large effect NOVA had on transcript (see *Table 1*) and protein levels in neurons, prompting us to examine whether additional NOVA actions on *Dlg3* intronic elements contributed to effects on steady-state RNA levels.

## *Dlg3* mRNA level is reduced in NOVA DKO through actions on cryptic NMD transcripts

The NOVA nuclear HITS-CLIP results demonstrated NOVA intronic and 3′ UTR binding sites that were present only in nuclear and cytoplasmic HITS-CLIP, respectively, (*Figure 2A*), suggesting a hierarchy of NOVA action. To explore whether NOVA intronic binding sites might relate to an action on *Dlg3* steady-state mRNA levels, we empirically searched for cryptic alternative splicing events in regions neighboring the largest intronic HITS-CLIP clusters, given previous observations indicating that NOVA intronic clusters predict alternative splicing outcomes (*Dredge et al., 2005*; *Ule and Darnell, 2006*; *Licatalosi et al., 2008*; *Zhang et al., 2010*). RT-PCR was performed in WT vs. NOVA DKO brain RNA using primers bounding introns harboring NOVA HITS-CLIP clusters (exons 14 (E14) and E18 in *Figure 2A*). In WT mouse brain two isoforms were detected and sequenced, and these were found to correspond to known alternative spliced isoforms, harboring what we refer to as E16, with or without E17, and a trace amount of a larger isoform. Remarkably, RT-PCR analysis of NOVA DKO mouse brain RNA revealed a marked increase of this larger isoform, and the presence of a previously unknown fourth,

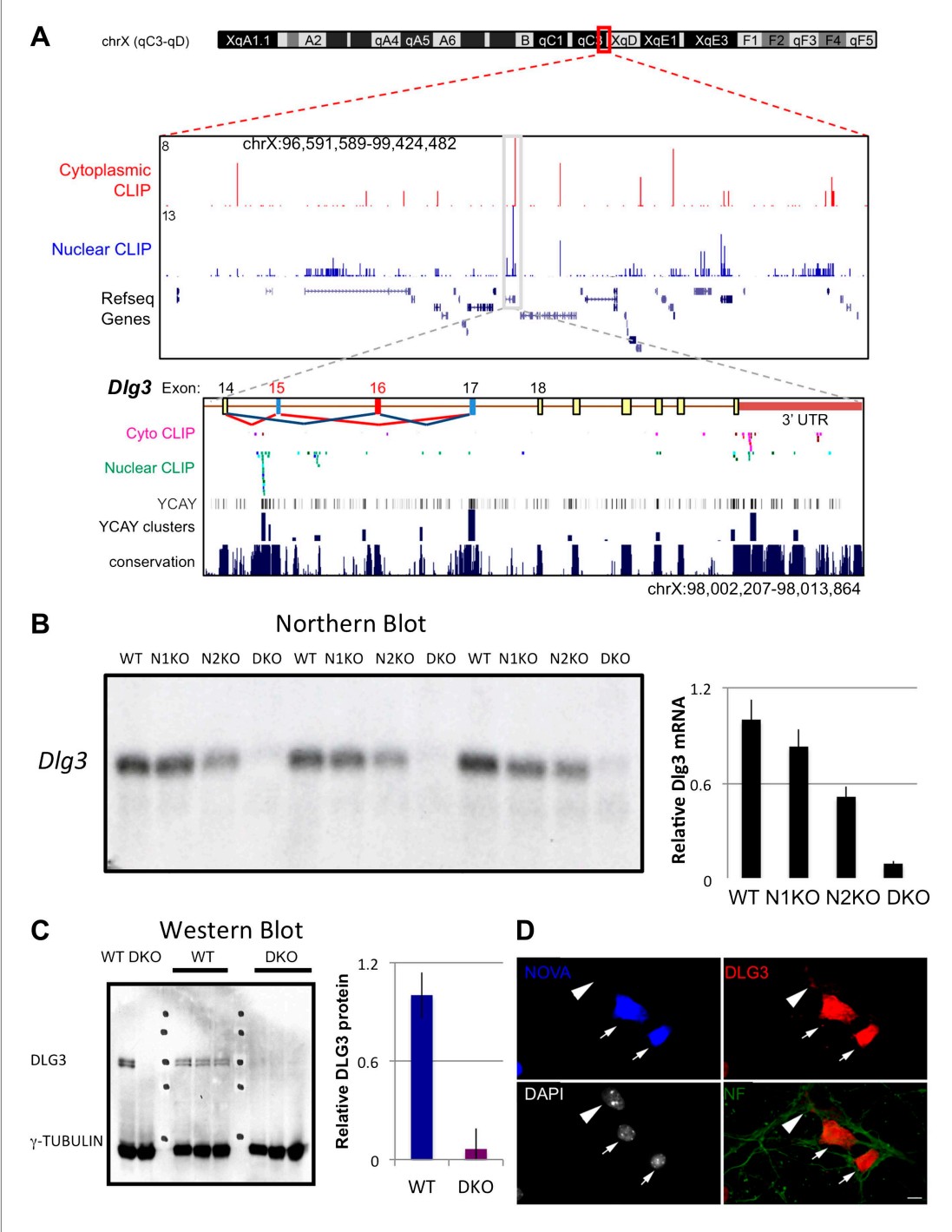

**Figure 2**. NOVA regulates the expression of *Dlg3* mRNA and protein. (**A**) Location of NOVA cytoplasmic and nuclear CLIP tags in chromosome X:96591589-99424482. Red and purple colors represent cytoplasmic CLIP tags and green and blue colors represent nuclear CLIP tags. The location of *Dlg3* is boxed in black and magnified in the lower box (chromosomeX:98002207-98013864). This higher magnification illustrates the position of *Dlg3* constitutive (yellow), alternative (colored) exons and 3′ UTR (brown) relative to CLIP tags, YCAY elements, and sequence conservation across species. More cytoplasmic tags were evident in the 3′ UTR and more nuclear tags in introns. Clusters of CLIP tags can be seen to overlap with the location of clusters of YCAY sequences (in grey) as well as bioinformatically predicated clusters of YCAY elements (in blue; see ***Zhang et al., 2010***). (**B**) Northern blot analysis of *Dlg3* mRNA from three biologic replicates of WT or Nova KO brain mRNA. Equal amount of RNA was loaded (see ***Figure 2—figure supplement 2***). Quantitation of relative RNA intensity (WT/DKO) was plotted as a relative ratio of *Dlg3* mRNA in WT, N1 KO, N2 KO or DKO brain as indicated; error bars represent standard deviation (p<0.05); about 90% of *Dlg3* mRNA is absent in DKO brain. (**C**) Immunoblot analysis of DLG3 in WT vs DKO. Protein

*Figure 2. Continued on next page*

*Figure 2. Continued*

extracts from the four different WT or DKO mouse brains (as indicated; E18.5) were assessed, and γ-TUBULIN was used as a normalizing control. Quantitation of protein intensity is indicated in graph to the right, plotted as relative ratio of DLG3 in WT/DKO, indicate that ~90% of DLG3 protein is absent in DKO brain; error bars represent standard deviation (p<0.05). (**D**) Immunofluorescence detection of DLG3 (red), NOVA (blue) and Neurofilament (NF) (green) proteins on WT/DKO mixed primary mouse neuronal cultures. DAPI and neurofilament stained all neuronal nuclei and processes, respectively, while NOVA staining differentiates WT and DKO neurons. The DLG3 signal was markedly reduced in DKO neurons. Scale bar: 10 µm.

The following figure supplements are available for figure 2:

**Figure supplement 1**. *Dlg3* mRNA isoforms in Nova KO brain.

**Figure supplement 2**. Northern blot analysis of *Dlg3* mRNA in Nova KO brain-reproducibility and control.

even larger form. Sequence analysis revealed the latter two isoforms correspond to the inclusion of an exon, E15, just downstream of the main Nova nuclear CLIP cluster, and a new combination of exons undetectable in WT brain, incorporating both E15 and E16 (as well as E17; termed Dlg3.15-16-17). E16 also has a second upstream intronic CLIP cluster, suggesting that the Nova binding sites upstream of E15 and E16 could normally act to suppress their dual inclusion, consistent with their inclusion in the DKO brain and with the Nova position-dependent splicing map (*Licatalosi et al., 2008*; *Licatalosi and Darnell, 2010*).

Interestingly, the previously unknown Dlg3 isoform (Dlg3.15-16-17) encoded a transcript harboring a PTC, formed by the inclusion of both E15 and E16 (see *Figure 3A*). This suggested that the marked reduction in DLG3 protein levels might relate to nonsense-mediated mRNA decay (NMD), an mRNA surveillance mechanism in which transcripts containing PTCs are recognized and degraded in association with early rounds of translation (*Schoenberg and Maquat, 2012*).

To assess whether NOVA acted on the Dlg3.15-16-17 through NMD, we examined the effect of blocking NMD on NOVA-dependent *Dlg3* splicing using multiple approaches (*Matsuda et al., 2008*). First, we added emetine, a translational inhibitor, into primary neuronal cultures of neurons obtained from WT or NOVA DKO mouse brain, since a pioneering round of translation is necessary to activate the NMD pathway (*Ni et al., 2007*; *Schoenberg and Maquat, 2012*). After emetine treatment for 10 hr, RNA was isolated and *Dlg3* splicing was assayed by RT-PCR. In WT neurons, emetine had no effect on *Dlg3* splicing. However, in the absence of NOVA, emetine induced a dramatic increase in the NMD isoform Dlg3.15-16-17, and total *Dlg3* mRNA levels increased by 3.5-fold (*Figure 3B*). This result establishes a link between the ability to detect NOVA-dependent splicing of Dlg3.15-16-17 and active translation, suggesting that this isoform is normally suppressed coordinately both by NOVA and NMD.

To more directly assess whether *Dlg3.15-16-17* is regulated by NMD, we knocked down a core component of the NMD pathway, Upf1 (*Mendell et al., 2002*), with Upf1 siRNA. We first established that Upf1 siRNA was able to reduce UPF1 protein (by ~90%) in N2a cells (*Figure 4—figure supplement 1*). We then transfected Upf1 siRNA into cultured primary neurons to readdress the possibility of coordinate regulation by NOVA and NMD. In primary neuronal cultures, transfection efficiency was lower, such that only about half of the cells were transfected, and we detected a concomitant ~50% decrease in *Upf1* transcript in qRT-PCR analysis (*Figure 4—figure supplement 2*). Hence to directly assess the effects of *Upf1* knock-down on a per-cell basis, we transfected *Upf1* siRNA into WT or DKO primary neuronal cultures, and monitored DLG3 and UPF1 protein levels in individual cells by immunofluorescence microscopy. These experiments revealed a reciprocal relationship between DLG3 protein and UPF1 protein levels in NOVA DKO neurons. DLG3 protein level was very low in untransfected DKO neurons that expressed UPF1 protein (see *Figures 2D and 3C*), whereas cells in which UPF1 signals were markedly reduced by *Upf1* siRNA exhibited a markedly increased DLG3 signal. We interpret these results to indicate that when the NMD pathway was inhibited by Upf1 knock-down, mis-spliced *Dlg3.15-16-17* mRNA led to accumulation of DLG3 protein that escape NMD degradation and accumulate at the protein level. This may have included truncated isoforms, as these experiments used an antibody against a Dlg3 N-terminal epitope; (see 'Materials and methods'). More generally, these results demonstrate that NOVA regulation of a cryptic alternatively spliced exon is necessary to prevent NMD-mediated suppression of Dlg3 expression in the brain, and this can act in concert with 3' UTR binding to maintain Dlg3 expression.

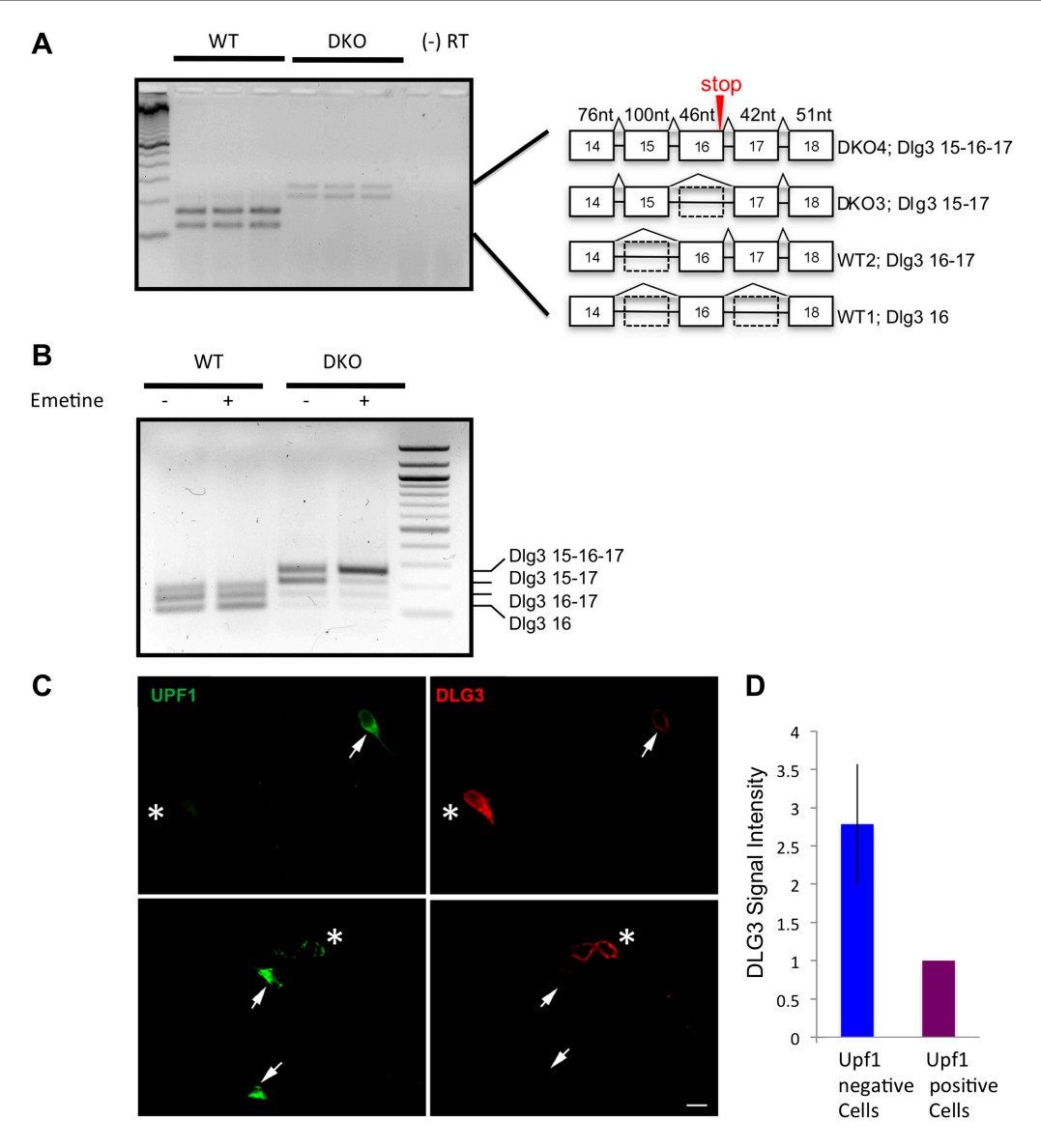

**Figure 3**. *Dlg3* mRNA is decreased through inclusion of cryptic NMD exons in the absence of NOVA. (**A**) RT-PCR and sequence analysis (*Figure 3—source data 1*) showed four *Dlg3* isoforms which have different combinations of alternatively spliced exons. The lower two PCR products (WT1 and WT2), respectively, harbor alternative exon 16 (E16) with or without E17, and both encode in-frame protein variants. The upper two PCR products, evident in DKO brain (DKO3 and DKO4), include E15, with or without E16, and include E17. The DKO4 isoform was not annotated in Refseq, and the combination of E15 and E16 led to a frameshift and inclusion of a premature stop codon (TAA) in E16, as indicated in the schematic (also leading us to color E16 red in *Figure 2A*). The 15–17 containing product does not make a premature stop codon. (**B**) After 6 DIV primary mouse neuronal cultures were treated with 100 µg/ml emetine for 10 hr, RNA was harvested from triplicate samples and analyzed by RT-PCR. Emetine treatment had no effect cell viability nor on isoforms produced in WT cells, but led to accumulation of the NMD isoform DKO4 in Nova DKO neurons. Spliced isoforms are indicated. (**C**) siRNA to Upf1 was transfected into DKO mouse primary cells, and after 24 hr, DLG3 (red) and UPF1 (green) protein was detected by immunofluorescence microscopy. Arrows indicate cells that have relatively high UPF1 levels; these cells have low DLG3 levels. In contrast, cells transfected with Upf1 siRNA (asterisks) had markedly reduced Upf1 levels and had increased (rescued) DLG3 levels. Upper and lower rows represent independent experiments. Scale bar: 10 µm. (**D**) Quantitation of DLG3 signal in Upf1 positive and Upf1 knockdown cells in (**C**). Signal intensity was normalized to the signal in Upf1 positive cells. Error bar represents standard deviation (p<0.05).

*Figure 3. Continued on next page*

*Figure 3. Continued*

The following source data and figure supplements are available for figure 3:

**Source data 1.** Sequence of RT-PCR products (from *Figures 3 and 5*).

**Figure supplement 1**. Reporter construct design for NOVA 3' UTR actions.

**Figure supplement 2**. NOVA increases the stability of some RNAs through binding to 3' UTR YCAY elements.

### *Scn9a* mRNA and protein level increases in NOVA DKO through NOVA-dependent splicing-coupled NMD

We next explored an example of the less common converse situation, in which microarray analysis revealed NOVA-dependent decreases in steady-state mRNA levels. We examined *Scn9a* mRNA, which showed the largest increase of 2.9-fold in NOVA DKO brain in the microarray data. We first independently validated this data, demonstrating that *Scn9a* mRNA level increased in NOVA DKO RNA by qRT-PCR (*Table 1*). We then determined whether the effects of NOVA on steady state *Scn9a* mRNA were reflected at the protein level. Western blot analysis revealed a marked reduction (~12-fold) of SCN9a protein in NOVA DKO brain (*Figure 4A,B*), indicating that NOVA normally acts to suppress SCN9a expression at both the RNA and protein levels in the brain.

When we examined *Scn9a* transcripts for NOVA binding in HITS-CLIP data, we found no 3' UTR tag clusters. Given that NOVA regulated *Dlg3* mRNA through repression of the NMD pathway, we explored whether an action of NOVA on NMD might also explain its effect on *Scn9a* mRNA steady-state levels. We found that *Scn9a* did harbor a major tag cluster in an intron upstream of a previously uncharacterized but conserved exon (*Figure 4D*; based on one sequenced transcript; NCBI accession: AM905322). Empirically, we assayed mRNA expression in this region from WT vs DKO brain by RT-PCR (*Figure 4C*). Interestingly, in WT brain the *Scn9a* transcript had two isoforms, but in NOVA DKO brain the larger isoform was lost. Sequence analysis demonstrated that this larger isoform corresponded to an alternative exon 17, termed E17a, that was not represented in RefSeq (as it corresponded to a partial mRNA sequence only), and that E17a harbored a PTC (*Figure 4D*). Again, this data was consistent with the upstream position of the NOVA CLIP tag clusters relative to E17a (*Figure 4C*) predicting (*Licatalosi et al., 2008*; *Licatalosi and Darnell, 2010*) that NOVA would normally act to suppress E17a inclusion.

These findings suggested that in the WT brain, Nova-mediated inclusion of *Scn9a* E17a was normally used to produce transcripts that are likely degraded by NMD, and that E17a exclusion in the absence of NOVA activity could lead to an increase in steady state mRNA and protein level. To test the role of NMD in NOVA-mediated regulation of *Scn9a* levels, we applied emetine to primary WT neurons to inhibit translation and NMD. After 10 hr, RT-PCR was performed using primers bounding E17a. Emetine treatment increased the levels of E17a-included *Scn9a* transcripts compared to non-treated cells (~75% increase; *Figure 4E*). In addition, we transfected Upf1 siRNA into WT primary neuronal cultures, and observed an increase in *Scn9a* E17a containing mRNA after siRNA KD (*Figure 4F*). Taken together, these observations indicate that *Scn9a* mRNA and consequently protein is held in balance by a NOVA-regulated NMD pathway.

### NOVA-dependent regulation through NMD

To find additional NOVA-dependent NMD target mRNAs, we searched HITS-CLIP data and exon array data for target RNAs that had unexplained NOVA intronic CLIP tags ('orphan clusters') and whose steady-state level was NOVA-dependent. Since not all alternative exons and splicing patterns are fully annotated, particularly those associated with NMD, we empirically searched for cryptic exons in introns that had robust NOVA orphan clusters in nuclear CLIP experiments. Among 15 transcripts showing significant NOVA-dependent changes (*Table 1*), 11 transcripts harbored robust NOVA intronic CLIP tag clusters. We designed RT-PCR primers in exons bounding these orphan intronic NOVA binding sites and assessed whether cryptic NOVA-dependent NMD exons could be identified. We identified and sequenced cryptic NOVA-regulated exons, evident only in NOVA DKO brain RNA, in 9 of these 11 transcripts, as well as in an additional 2 transcripts (see *Table 1*, and *Figure 3—source data 1*), and found PTCs in each (*Figure 5—figure supplement 1*). In 7 transcripts splicing was consistent with NOVA-dependent inclusion of an NMD exon, such that these exons were absent in NOVA DKO brain,

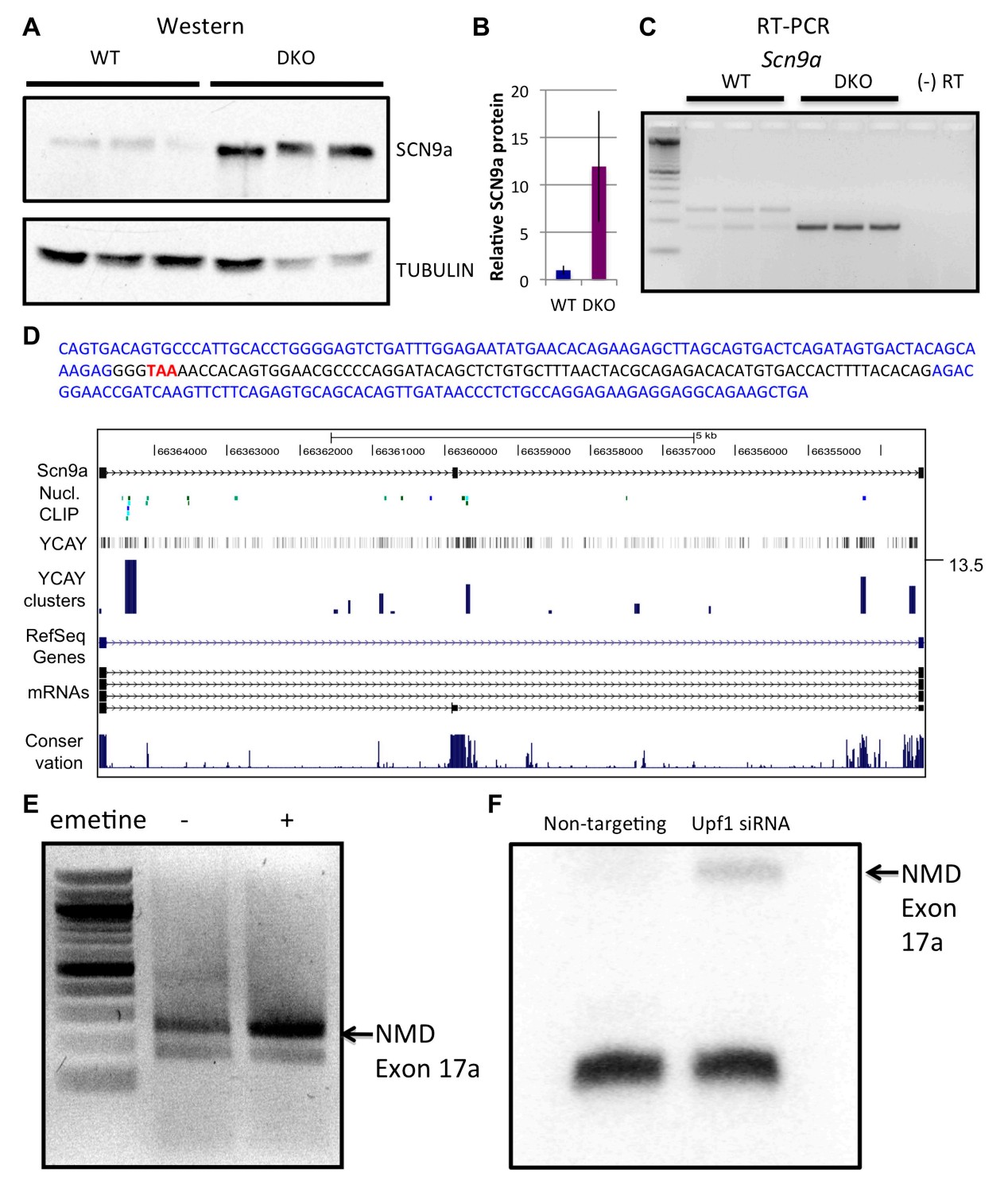

**Figure 4**. NOVA and NMD-mediated regulation of *Scn9a* mRNA and protein. (**A**) Immunoblot analysis of SCN9a in WT vs DKO. Proteins extracts from WT (lane 1–3) vs DKO (lane 4–6) were loaded. γ-Tubulin was used as a normalizing control. (**B**) Quantitation of relative protein intensity (WT/DKO). The results were plotted as a relative ratio of SCN9a in WT/DKO; error bars represent standard deviation (p<0.05). About 90% signal was reduced in WT. (**C**) RT-PCR from WT and DKO (three biologic replicates) shows NOVA-dependent splicing of *Scn9a*. WT brains express two alternative splicing isoforms (lanes 1–3), while DKO brains express only the smaller isoform (lanes 4–6). A (−) RT control is indicated; primers are given in **Supplementary file 2**. (**D**) Sequence analysis and map of the spliced isoforms from Figure 4C showed that the larger band corresponds to a transcript in which an exon (17a) was included,
*Figure 4. Continued on next page*

*Figure 4. Continued*
introducing a premature stop codon. In the sequence shown, exon 17a is highlighted in black, and the TAA premature stop codon is indicated in red. (**E**) Six DIV WT primary mouse neuronal cultures were treated with emetine, as indicated, for 10 hr and RT-PCR was performed. The NMD exon was increased after emetine treatment. (**F**) siRNA to *Upf1* or a non-targeting siRNA were transfected in WT mouse primary cells, and RT-PCR was performed with the same primers used in Figures 4C,E. The intensity of the NMD exon was increased specifically after *Upf1* siRNA treatment.

The following figure supplements are available for figure 4:

**Figure supplement 1**. N2A cells: Upf1 siRNA efficiently reduces endogenous UPF1 protein levels.

**Figure supplement 2**. Primary neurons: Upf1 siRNA reduces endogenous UPF1 protein levels.

and in each case these correlated with increased steady-state mRNA levels in NOVA DKO brain. Conversely, in 4 transcripts splicing of these cryptic exons was consistent with NOVA-dependent suppression of an NMD exon and decreased steady-state levels in NOVA DKO brain.

To assess whether these transcripts were indeed regulated through NMD, we added emetine to primary neuronal cultures and examined the steady-state expression of the putative NMD-exons. After emetine treatment for 10 hr, qRT-PCR was performed on five transcripts in WT and DKO neurons. When we plotted normalized fold changes on a $\log_2$ scale, the results demonstrated that *Dlg3*, Rasgrf1 and Slc4a3 were degraded by NMD in DKO relative to WT mice, while Stx2 and *Scn9a* degradation by NMD in WT mice was alleviated in NOVA DKO animals (*Figure 5B*; *Figure 5—figure supplement 2*). Taken together, these data indicate the extensive use of NOVA in steady-state neurons to inhibit cryptic NMD exons, and, to a lesser degree, mediate inclusion of NMD exons, as a means to maintain or reduce, respectively, the steady-state levels of neuronal transcripts and proteins.

## Dynamic control of NOVA-regulated NMD exons

To assess whether NOVA-dependent regulation of NMD exons is dynamic and physiologically relevant, we examined the effects of inducing seizures in vivo on the utilization of these alternative exons. Mice were treated with pilocarpine or sham to induce acute generalized seizures, using previously established protocols (*Turski et al., 1986*). We assayed activity-dependent inclusion of NMD exons in *Scn9a*, *Cdk5rap2* and *Stx2*, which we had found to be included by NOVA in the resting brain. In each transcript, pilocarpine induced seizures correlated with clear increases in utilization of the NMD exon (*Figure 6A,B*), suggesting that the levels of these transcripts can be dynamically altered by electrical activity altering NOVA action.

Pilocarpine can induce neuronal loss days after seizure (*Turski et al., 1986*), although this is unlikely to be the case in these experiments, as RNA was analyzed 2 hr after seizure, and direct examination of neurons by immunofluorescence microscopy gave no indication of neuronal loss (see below, *Figure 7A*). Nonetheless, to assess whether this protocol, which induces neuronal stress, affects splicing in a global or specific manner, we examined global and Nova-dependent splicing changes at various times after pilocarpine induced seizures (*Table 3*).

We first examined global changes in splicing using exon junction arrays at 4 hr and 24 hr after induction of pilocarpine seizure. Remarkably, there were very few splicing changes detected in the brain after pilocarpine seizure. Using stringent criteria ($|\Delta I| > 0.2$), only 23 exons showed differential splicing 24 hr after pilocarpine treatment, with more changes (96) evident 4 hr after pilocarpine treatment (*Supplementary file 1*); by contrast, using the same stringency, 1,239 exons showed differential splicing in brain relative to thymus tissue ($|\Delta I| > 0.2$; (*Ule et al., 2005b*)). These results indicate that the regulation of alternative splicing is not globally perturbed within 24 hr of epileptic seizure in the mouse brain.

We validated exons that did show seizure-induced splicing changes, and examined their relationship to Nova-regulated exons. We measured the ratio of alternate exon inclusion:exclusion by RT-PCR, validating 35 targets at 4 hr and 24 hr time points (*Supplementary file 1*). GO analysis revealed that transcripts showing these changes showed a 7.5-fold enrichment of synaptic proteins, and a 5.2-fold enrichment for proteins with serine/threonine kinase activity, including SNAP25 and BDNF, proteins shown to play important roles during synaptic activity (*Yoshii and Constantine-Paton, 2010*; *Kennedy and Ehlers, 2011*). We have previously found that NOVA regulates ~800 alternate splicing events, about half of which are cassette exons (*Zhang et al., 2010*). We examined the 35 alternate exons that showed pilocarpine-dependent changes to see

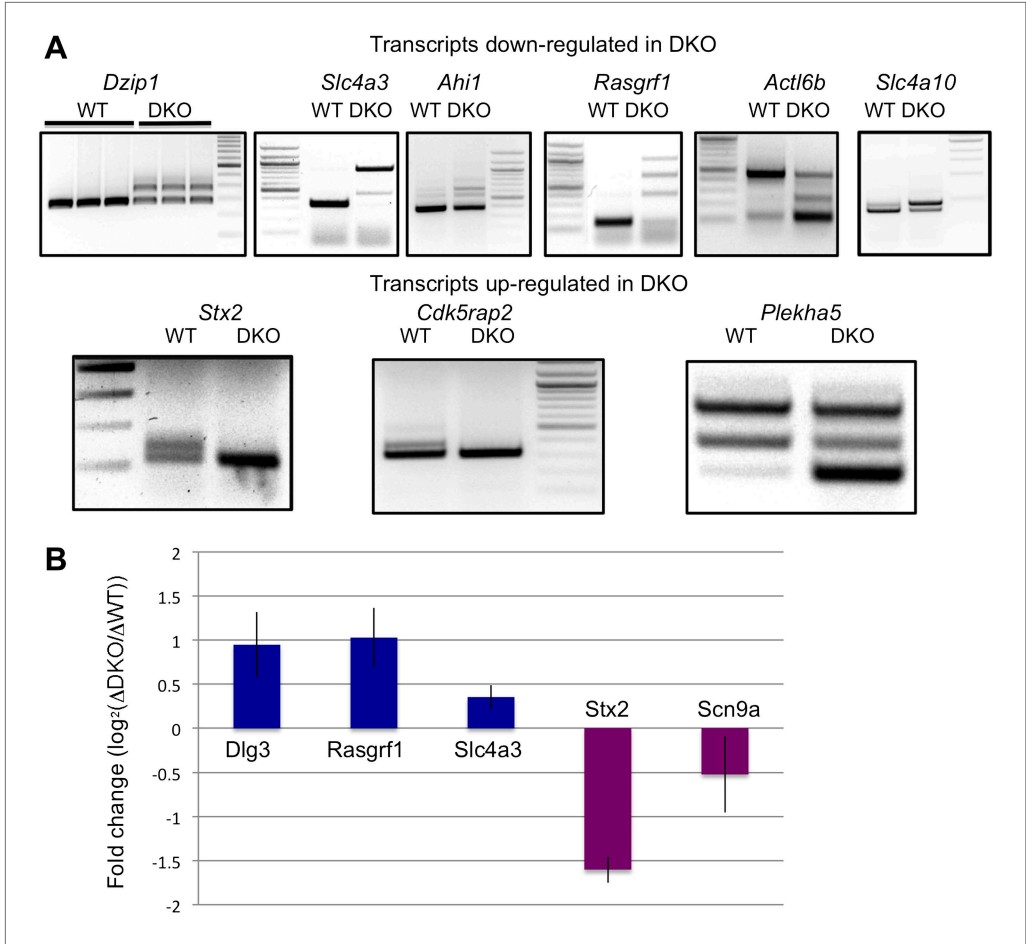

Figure 5. NOVA regulates cryptic NMD exons and transcript levels. (**A**) Analysis of alternative spliced isoforms in transcripts chosen solely on exon array data showing NOVA-dependent steady-state mRNA changes and robust HITS-CLIP clusters in introns. Transcripts were then screened for the presence of cryptic NMD exons by RT-PCR using primers in exons bounding the intronic HITS-CLIP clusters. Data is divided into those transcripts down-regulated or up-regulated in Nova DKO, as indicated. Sequence analysis of RT-PCR products showed the presence of cryptic exons harboring premature stop codons (**Figure 3—source data 1**; **Supplementary file 2**). A diagram of the loci of each NMD exon present in Figure 5A is shown in **Figure 5—figure supplement 1**. For example, most transcripts down-regulated in Nova DKO brains show a larger, PTC containing exon in DKO; one exception is Actl6b, in which in the absence of NOVA there is a PTC, and in WT brain, an upper alternate isoform (exon) is present that corrects that frame-shift; (**B**) Effect of emetine on putative NOVA-regulated cryptic NMD exons. The steady-state level of six transcripts identified in **Figures 3B, 4C and 5A** were assessed by qRT-PCR in six DIV WT vs Nova DKO primary mouse neurons incubated for 10 hr in the presence or absence of emetine. The results were plotted with the Y-axis as a measure of the degree of putative NOVA-dependent NMD regulation (the fold change of transcript levels in DKO neurons in the presence or absence of emetine, divided by that of WT, in $\log_2$ scale). For example, for *Dlg3* the $\log_2$ value is about 1.0 indicating that emetine treatment increased the *Dlg3* NMD-isoform in DKO neurons relative to WT neurons by a factor of two, while emetine decreased the NMD isoform of *Scn9a* by ~1.4-fold, leading to decrease or increase in the respective proteins in Nova DKO neurons (**Figures 2 and 3** or **Figure 4**, respectively). Three independent experiments were performed and error bars represent standard deviation (p<0.05).

The following figure supplements are available for figure 5:

**Figure supplement 1**. Diagrams of each of the NMD exons shown in **Figure 5A**

**Figure supplement 2**. NOVA regulates the expression of Stx2 (Syntaxin 2) mRNA and protein.

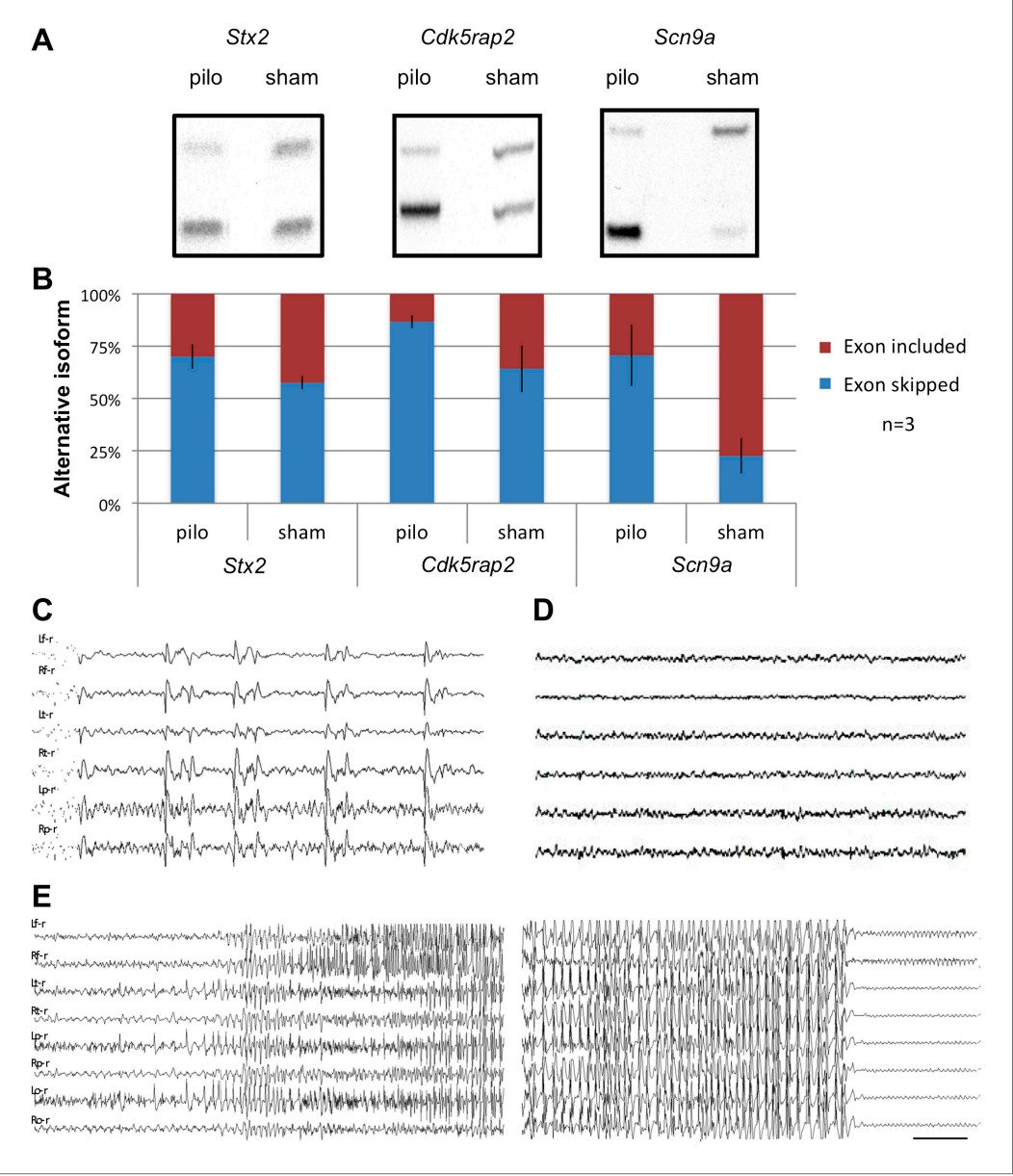

**Figure 6**. Seizure-induced inhibition of NOVA-regulated NMD exons. (**A**) Mice were treated with pilocarpine (pilo) to induce seizures or were mock-treated (sham). 2 hr later brains were harvested and splicing of NMD exons assayed by RT-PCR. (**B**) Quantitation of experiments described in (**A**), from three biologic replicates. Error bars represent standard deviation (p<0.05; Student's t-test). (**C**),(**D**) EEG of freely moving Nova2$^{+/-}$ mutant displays frequent synchronous cortical interictal discharges (**C**) not detected in wild type mice (**D**). (**E**) Spontaneous generalized seizure discharge in adult Nova2$^{+/-}$ mutant. A 20 s gap of continuous hypersynchronized EEG pattern separates the beginning and end of the seizure discharge. Bilateral left and right frontal, temporal, parietal (**C**, **D**) and occipital (**E**) leads are shown. Time calibration 1 s (**C**, **D**), 2 s (**E**).

The following figure supplements are available for figure 6:

**Figure supplement 1**. Comparison of different sets of transcripts regulated by NOVA.

which were present on our list of Nova-dependent exons, and identified only eight as Nova-regulated (*Supplementary file 1*). These data indicate that only a small fraction of potential regulated alternate exons, including those regulated by Nova, are affected by seizure activity, and that pilocarpine does not appear to have a global impact on Nova splicing activity.

**Table 3.** Splicing changes in brain after pilocarpine-induced seizures

**|ΔI t-test| ≥ 5.6 (p<0.005)**

| 4 hr | Total | Validated |
|---|---|---|
| \|ΔI t-test\| ≥ 0.4 | 7 | 2 |
| \|ΔI t-test\| ≥ 0.3 | 21 | 5 |
| \|ΔI t-test\| ≥ 0.2 | 117 | 22 |

**|ΔI t-test| ≥ 8.6 (p<0.001)**

| 4 hr | Total | Validated |
|---|---|---|
| \|ΔI t-test\| ≥ 0.4 | 7 | 2 |
| \|ΔI t-test\| ≥ 0.3 | 18 | 5 |
| \|ΔI t-test\| ≥ 0.2 | 96 | 22 |

**|ΔI| ≥ 0.15**

| 24 hr | Total | Validated |
|---|---|---|
| \|ΔI\| ≥ 0.3 | 3 | 1 |
| \|ΔI\| ≥ 0.2 | 23 | 6 |
| \|ΔI\| ≥ 0.15 | 56 | 6 |

Exon junction array results assessing splicing changes in the hippocampus after pilocarpine seizure, compared to sham controls. The total number of alternative exons identified from the brains of pilocarpine treated animals from which RNA was analyzed by genome-wide exon junction arrays is shown. The data in is organized as a function of differing stringency thresholds (as previously described: ΔI, a measure of the inclusion of exons in sham relative to pilocarpine treated animals; ΔI was determined using the ASPIRE algorithm (*Ule et al., 2005b*) and ΔI t-test using ASPIRE2 (*Licatalosi et al., 2008*)). Each data point represents analysis of all results from three pairs of biological replicates. Validated refers to independent qRT-PCR validation (see *Supplementary file 1*) of selected transcripts from total.

## Cortical hyperexcitability and epilepsy in Nova2 heterozygous mice

To further assess the relationship between Nova and epilepsy, we examined EEGs in Nova-null mice. However, Nova-null mice die by 2–3 weeks of birth, and we instead examined Nova-2 heterozygotes, which show partial defects (~50% or less) changes in a number of NOVA regulated alternative exons (*Ule et al., 2005b*). Background cortical activity recorded in heterozygous mutants showed frequent bilateral synchronous cortical epileptiform discharges (*Figure 6C*) that were not seen in control +/+ mice (*Figure 6D*). A second abnormality observed was the spontaneous occurrence of generalized seizures (*Figure 6E*). Thus perturbing NOVA steady-state levels in vivo has relevant physiologic consequences, as haploinsufficiency is sufficient to give rise to spontaneous epilepsy in mice.

## Activity-dependent changes in NOVA subcellular localization

To further explore the relationship between electrical activity and NOVA action on splicing, we asked whether we could detect any acute effect of pilocarpine seizure on NOVA itself. We have previously demonstrated that NOVA shuttles between the nucleus and cytoplasm (*Racca et al., 2010*), and hence explored the possibility that electrical activity might affect the cellular distribution of the protein. We examined NOVA subcellular localization by immunofluorescence microscopy after sham or pilocarpine induced seizures. We consistently observed marked shifts of NOVA from the neuronal nucleus to the cytoplasm within 2–4 hr of pilocarpine-induced seizures, and that persisted in some animals for 24 hr (*Figure 7A*). This finding was evident in cortical neurons, and even more so in the CA1 pyramidal neurons of the hippocampus, where quantitation revealed over a 6-fold shift in the ratio of nuclear to cytoplasmic signal intensity at 4 hr after seizure (*Figure 7B*). Consistent with the cortical nature of epilepsy, we did not see this shift in NOVA localization in Purkinje neurons of the cerebellum (*Figure 7A*). These data suggest that there may in part be cell biologic mechanisms whereby pilocarpine affects the ability of NOVA to regulate splicing of some targets, but additional experiments will be required to fully understand their nature and consequences.

## NOVA regulates some transcript levels by binding the 3′ UTR

To further address the role NOVA plays in 3′ UTR regulation of mRNA steady-state levels, we searched for transcripts with a high percentage of 3′ UTR tags. We found that Synaptogyrin III (*Syngr3*), Glycine receptor B (*Glrb*) and the Gaba-B receptor1 (*Gabbr1*) had robust 3′ UTR CLIP tag clusters (*Figure 8A*). RNA levels quantitated by RT-PCR in WT or DKO brain demonstrated reduced levels of all three mRNAs in NOVA DKO (*Figure 8B*). Decrease of *Syngr3* mRNA in NOVA DKO was confirmed by Northern blot (~20% reduction; *Figure 8C,D*) and at the protein level by Western blot (~35% reduction *Figure 8E,F*).

To address whether direct NOVA binding to the *Syngr3* or other 3′ UTR (YCAY) elements could stabilize mRNA, we generated reporter constructs in which EGFP was generated from a transcript harboring WT or YAAY-mutated NOVA 3′ UTR binding sites defined by CLIP (*Figure 3—figure supplement 1*).

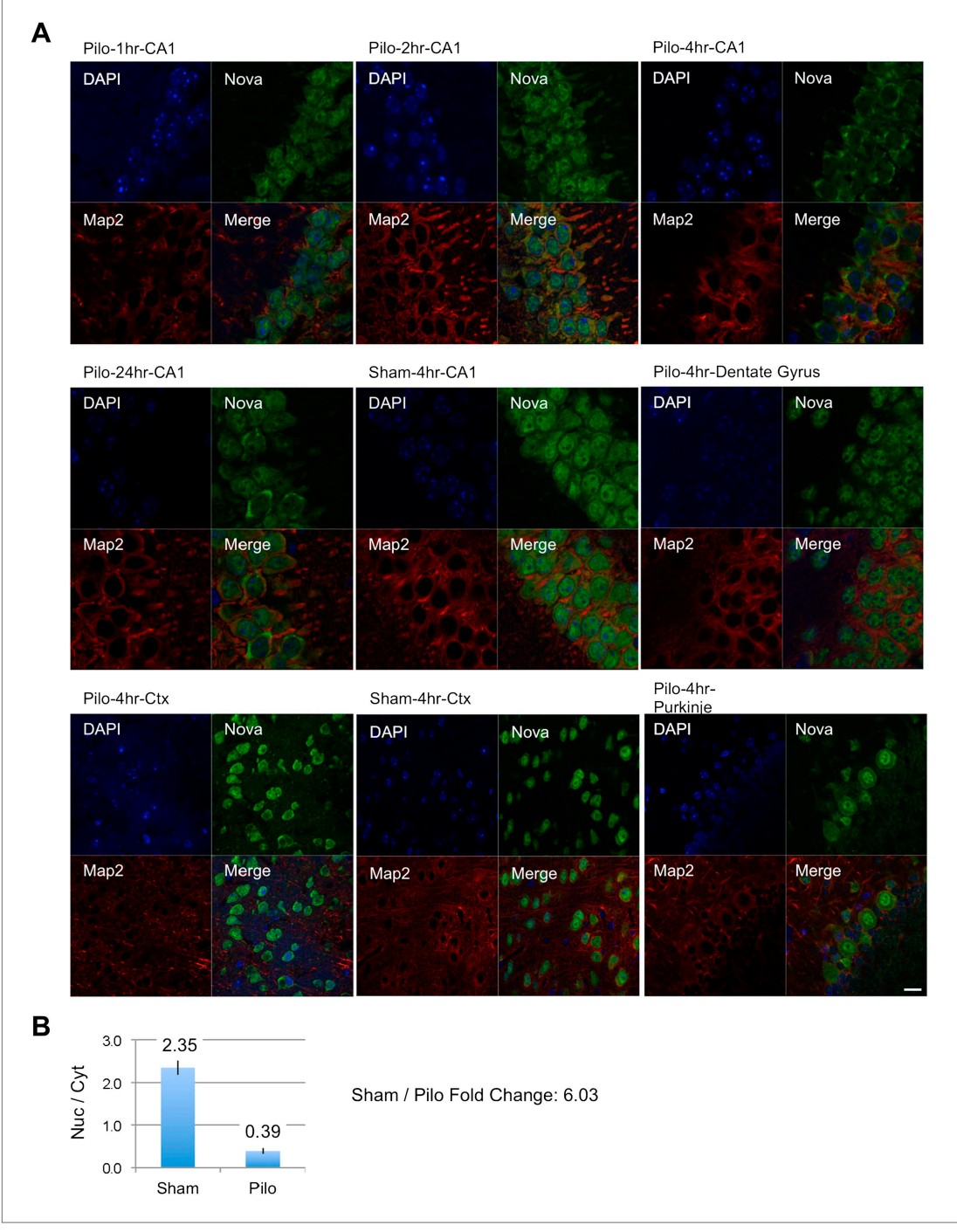

**Figure 7**. NOVA subcellular localization after pilocarpine-induced status epilepticus. (**A**) NOVA and MAP2 protein localization were visualized by immunofluorescence (IF) in mouse brain sections from pilocarpine treated (pilo) or control animals (sham). Regions include CA1 and dentate gyrus (DG) from hippocampal area, cortex (Ctx) and cerebellar Purkinje neurons. Changes were clearly evident in CA1 neurons, were not evident in DG, and were variable in Ctx. No changes were expected nor observed in Purkinje neurons. (**B**) Quantification of IF signal intensity from nuclear area (Nuc) divided by signal intensity from cytoplasmic area (Cyt) in CA1 neurons. Nuc/Cyt ratios (sham divided by pilocarpine signal) were obtained from 22 cells in two sham animals and 23 cells in four pilocarpine animals at the 4 hr time point. Scale bar 20 μm.

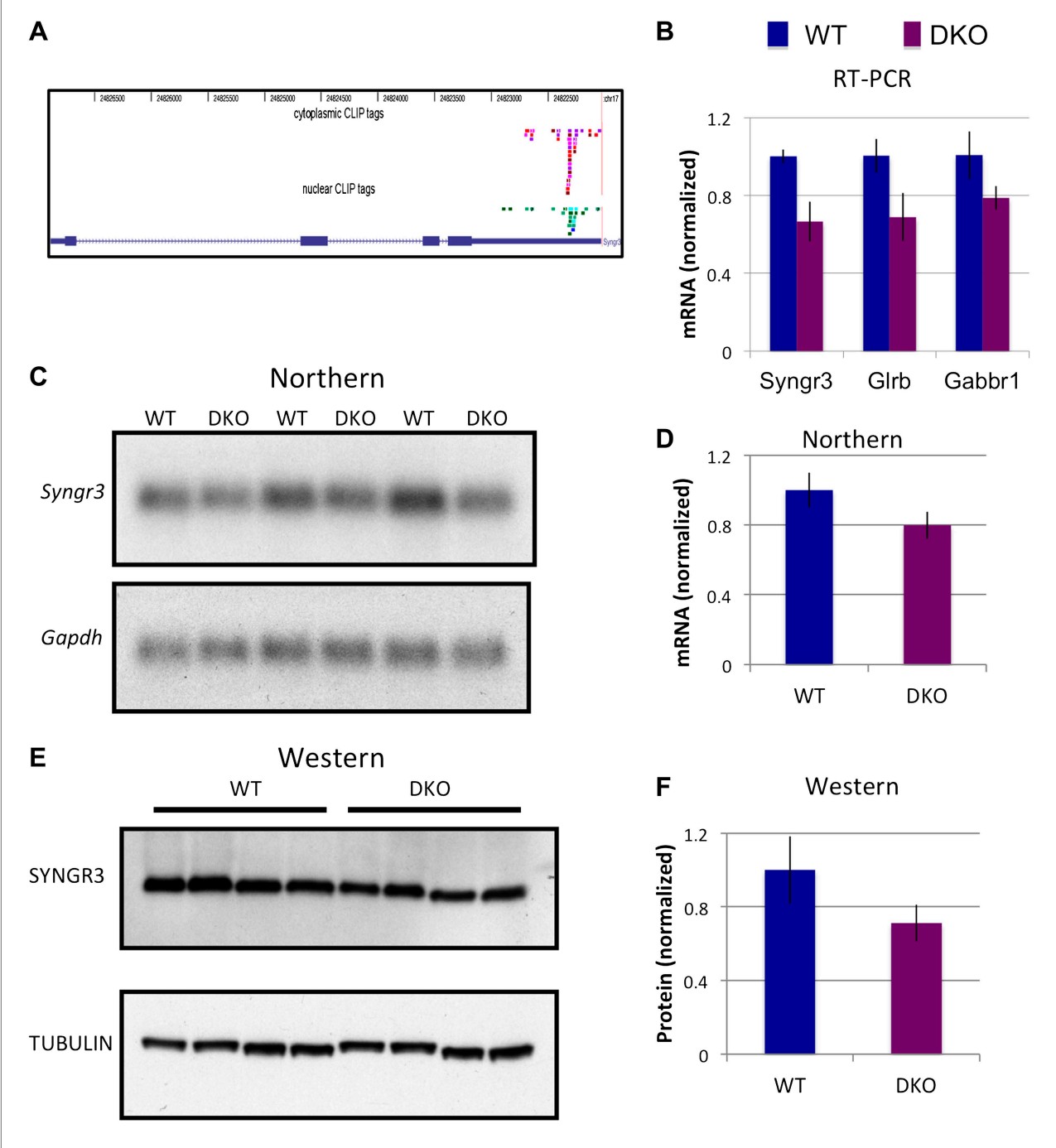

**Figure 8**. NOVA regulates the expression of synaptogyrin III (Syngr3) mRNA and protein. (**A**) Location of cytoplasmic and nuclear CLIP for *Syngr3*. Red and purple colors represent Cytoplasmic CLIP tags and green and blue tags represent nuclear CLIP tags. All tags were located in 3′ UTR. (**B**) qRT-PCR data of *Syngr3*, *Glrb* (Glycine receptor b), and *Gabbr1* (GABA B receptor1) showed mild reduction (about 30%) in *Nova* DKO brain. Y-axis represents the relative mRNA levels (WT/DKO). p<0.05. (**C**) Northern blot analysis of Syngr3 mRNA in WT (lane 1, 3, 5) vs DKO (lane 2, 4, 6). Gapdh probe was used as a normalizing control. (**D**) Quantitation of relative RNA intensity (WT/DKO) was plotted as a relative ratio of Syngr3 mRNA/Gapdh in *Nova* WT/DKO; error bars represent standard deviation (p<0.05); Syngr3 was reduced by about ~ 20% in DKO brain. (**E**) Immunoblot analysis of SYNGR3 in WT vs DKO. Protein extracts from WT (lane 1–4) vs DKO (lane 5–8) were loaded. γ-TUBULIN is used as a normalizing control. (**F**) Quantitation of relative protein intensity (WT/DKO). The results were plotted as relative ratio of SYNGR3 in WT/DKO; error bars represent standard deviation (p<0.05); SYNGR3 protein was reduced ~35% in DKO brain.

After transfecting each construct into N2a cell lines, cells were treated with actinomycin D for 5 hr to block new RNA synthesis, and qRT-PCR was performed to measure mRNA levels of each construct. These data showed that transcripts containing 3′ UTR YCAY elements were stabilized relative to those harboring YAAY elements (by a factor of 25–65%) compared to YAAY elements (*Figure 3—figure supplement 2*). Taken together (*Figure 8* and *Figure 3—figure supplement 2*), these data suggest NOVA protein is able to regulate mRNA levels through direct binding to 3′ UTR YCAY elements.

## NOVA-regulated RNAs are enriched in synaptic function related genes

To investigate whether transcripts whose steady-state levels are NOVA-regulated are more enriched in certain functional classes of genes than others, we analyzed the functional annotations associated with 229 genes from exon array data (*Figure 1A*) using GO analysis (*Table 4*). We found that these transcripts encode proteins that regulate synaptic biology or transport. We compared this set of transcripts with those previously found to be regulated by NOVA at the level of alternative splicing but without changes in steady-state mRNA levels. Interestingly, while both sets of transcripts encode proteins related to synaptic function, only 18 genes overlap between the two sets (*Figure 6—figure supplement 1*). This implies that the set of functions NOVA regulates at steady-state level differ significantly from those functions NOVA regulated strictly at the level of alternative splicing.

## Discussion

Regulation of steady-state mRNA levels plays a key role in cellular and neuronal plasticity, allowing tunable and dynamic regulation of protein output. Such regulation is generally thought to be mediated by RNA–protein interactions in the 3′ UTR, which may involve interactions with a number of RNA binding proteins. For example, the balance of binding to AU-rich elements by AUF1, BRF1, KSRP, TTP or HuR (*Meisner and Filipowicz, 2011*), binding to UGUR elements by Pumillio (*Crittenden et al., 2002*; *Quenault et al., 2011*), or interactions between miRNAs and the Ago–RISC complex (*Krol et al., 2010*) all feed into pathways of deadenylation and mRNA decay. Our finding that transcripts showing NOVA-dependent steady-state regulation did not have NOVA binding sites in the 3′ UTR was therefore unexpected, and might have suggested that these were indirect targets of NOVA action. However, HITS-CLIP provides a powerful means of providing unambiguous and unbiased sites of RNA–protein interaction. Directed analysis of unbiased HITS-CLIP targets can reveal cryptic biology, as in the analysis of developmental HITS-CLIP maps to study NOVA interactions with transcripts of the Reelin pathway and thereby identify a previously unknown NOVA-dependent alternative exon necessary for proper neuronal migration (*Yano et al., 2010*). Here, we have used the predictive power of HITS-CLIP to identify seemingly irrelevant intronic NOVA binding sites on transcripts showing unexplained NOVA-dependent steady-state changes to predict the presence of cryptic alternative NMD exons.

## NOVA-regulated splicing controls mRNA levels by NMD

NMD is an RNA surveillance mechanism that degrades mRNAs harboring premature termination codons (PTC), and is generally thought of as a means to protect cells from inadvertently producing truncated proteins that could be harmful to the cell. Such transcripts harbor downstream exon junction complex (EJC) proteins that, due to upstream PTCs, fail to be removed during an initial round of translation, leading to recruitment of NMD components (*Schoenberg and Maquat, 2012*). Recent studies have extended this view, indicating that NMD can function beyond a quality control mechanism, to serve as a mechanism for regulating gene expression disability (*Huang and Wilkinson, 2012*). NMD efficiency varies in different tissues, loss-of-function mutations in Upf genes in mice alter expression of a subset of transcripts, and loss-of-function mutations in Upf3b in humans are associated with intellectual disability (*Tarpey et al., 2007*).

Regulated alternative splicing of NMD exons has also been shown to act as a mechanism to regulate transcript levels, as in the case of autoregulation of splicing factors such as SC35, Ptbp1 and 9G8 (*Lewis et al., 2003*), as well as core components of the spliceosome (*Saltzman et al., 2008*). In addition, alternative splicing-coupled NMD has been shown to regulate expression levels of Arc (*Giorgi et al., 2007*), PSD95 (*Zheng et al., 2012*), and spermine acetyltransferase (*Hyvönen et al., 2006*), a process referred to as regulated unproductive splicing and translation (RUST) (*Lewis et al., 2003*). In yeast, a direct role for NMD was statistically inferred in the regulation of hundreds of transcripts involved in chromosomal and telomere maintenance and silencing (*Guan et al., 2006*). Cancer cells

**Table 4.** GO analysis

| Term | Count | % | p-Value | Fold | Benjamini |
|---|---|---|---|---|---|
| Biological process | | | | | |
| Transport | 57 | 27.67% | $1.40 \times 10^{-08}$ | 2.10779601 | $7.27 \times 10^{-05}$ |
| Establishment of localization | 57 | 27.67% | $3.74 \times 10^{-08}$ | 2.0512741 | $9.71 \times 10^{-05}$ |
| Localization | 60 | 29.13% | $1.46 \times 10^{-07}$ | 1.91908255 | $2.53 \times 10^{-04}$ |
| Ion transport | 25 | 12.14% | $1.50 \times 10^{-06}$ | 3.08118005 | 0.001943 |
| Synaptic transmission | 12 | 5.83% | $1.16 \times 10^{-05}$ | 5.49530653 | 0.011935 |
| Chloride transport | 7 | 3.40% | $4.92 \times 10^{-05}$ | 10.5684476 | 0.041641 |
| Transmission of nerve impulse | 12 | 5.83% | $6.38 \times 10^{-05}$ | 4.56499873 | 0.04624 |
| Exocytosis | 8 | 3.88% | $6.71 \times 10^{-05}$ | 7.88782093 | 0.042589 |
| Cation transport | 17 | 8.25% | $1.41 \times 10^{-04}$ | 3.04192354 | 0.07808 |
| Metal ion transport | 15 | 7.28% | $1.72 \times 10^{-04}$ | 3.30910296 | 0.085535 |
| Cell–cell signaling | 13 | 6.31% | $4.67 \times 10^{-04}$ | 3.37670829 | 0.197738 |
| Cellular component | | | | | |
| Synapse | 19 | 9.22% | $1.69 \times 10^{-10}$ | 7.12724905 | $1.33 \times 10^{-07}$ |
| Postsynaptic membrane | 12 | 5.83% | $3.96 \times 10^{-08}$ | 9.73280098 | $1.55 \times 10^{-05}$ |
| Synapse part | 12 | 5.83% | $1.25 \times 10^{-07}$ | 8.71242669 | $3.27 \times 10^{-05}$ |
| Cytoplasmic part | 66 | 32.04% | $1.11 \times 10^{-06}$ | 1.75847144 | $2.17 \times 10^{-04}$ |
| Cell junction | 18 | 8.74% | $2.18 \times 10^{-06}$ | 4.05127841 | $3.42 \times 10^{-04}$ |
| Cytoplasm | 91 | 44.17% | $1.52 \times 10^{-05}$ | 1.44439091 | 0.00198 |
| Plasma membrane | 44 | 21.36% | $5.11 \times 10^{-05}$ | 1.85625586 | 0.005706 |
| Molecular function | | | | | |
| Ion transmembrane transporter activity | 30 | 14.56% | $1.74 \times 10^{-10}$ | 4.09234024 | $4.70 \times 10^{-07}$ |
| Transporter activity | 42 | 20.39% | $4.86 \times 10^{-10}$ | 2.91474846 | $6.58 \times 10^{-07}$ |
| Substrate-specific transmembrane transporter | 31 | 15.05% | $1.15 \times 10^{-09}$ | 3.65758191 | $1.04 \times 10^{-06}$ |
| Transmembrane transporter activity | 33 | 16.02% | $2.22 \times 10^{-09}$ | 3.36413152 | $1.50 \times 10^{-06}$ |
| Substrate-specific transporter activity | 33 | 16.02% | $1.85 \times 10^{-08}$ | 3.08048906 | $9.99 \times 10^{-06}$ |
| Gated channel activity | 17 | 8.25% | $3.63 \times 10^{-08}$ | 5.82250711 | $1.64 \times 10^{-05}$ |
| Ligand-gated channel activity | 11 | 5.34% | $5.03 \times 10^{-08}$ | 11.1421945 | $1.70 \times 10^{-05}$ |
| Ligand-gated ion channel activity | 11 | 5.34% | $5.03 \times 10^{-08}$ | 11.1421945 | $1.70 \times 10^{-05}$ |
| Anion transmembrane transporter activity | 12 | 5.83% | $6.01 \times 10^{-08}$ | 9.36542127 | $1.81 \times 10^{-05}$ |
| Ion channel activity | 18 | 8.74% | $2.53 \times 10^{-07}$ | 4.7344533 | $6.84 \times 10^{-05}$ |
| Substrate specific channel activity | 18 | 8.74% | $3.85 \times 10^{-07}$ | 4.59483135 | $9.47 \times 10^{-05}$ |
| GO term analysis (top increase in DKO 14 genes) | | | | | |
| Biological process | | | | | |
| Vesicle-mediated transport | 3 | 21.43% | 0.02265 | 11.0206034 | 35.4494 |
| Molecular function | | | | | |
| Binding | 11 | 78.57% | 0.031126 | 1.41461519 | 1 |
| Transporter activity | 4 | 28.57% | 0.045036 | 4.34057779 | 1 |
| Protein binding | 9 | 64.29% | 0.005192 | 2.31262748 | 0.999999 |

GO (gene ontology) term analysis of transcripts up/down-regulated in WT vs DKO (see **Figure 1A**). Data of upper table are derived from top 215 down-regulated transcripts in DKO; (positive value in **Figure 1A**). Data of lower table are from top 14 up-regulated transcripts in DKO; (negative value in **Figure 1A**). Taken together, two tables suggested that NOVA regulates the levels of many transcripts relating to synaptic transmission (**Dennis et al., 2003**; **Huang et al., 2009**).

have been shown to utilize NMD, for example via actions on mutant cadherin transcripts (*Chd1*) that are associated with poor clinical outcome in hereditary diffuse gastric cancer. A full understanding of such pathways and means to manipulate them has been challenging because the factors regulating NMD-associated splicing are not generally known. Some well-studied exceptions have provided significant biologic insights, such as the autoregulation of the splicing factor PTBP1 (*Wollerton et al., 2004*) and its action on the paralogous transcript PTBP2, which mediates skipping of Ptbp2 exon 10, leading to NMD and suppression of neuronal differentiation (*Boutz et al., 2007*; *Makeyev et al., 2007*). Indeed, Makeyev and co-workers have identified PTBP1-dependent retained introns that suppress expression of neuronal transcripts in non-neuronal cells, allowing their expression as PTBP1 is downregulated during neuronal differentiation (*Yap et al., 2012*). We recently found that PTBP2 is expressed in neuronal progenitors, while NOVA is induced in post-mitotic neurons (*Yano et al., 2010*; *Licatalosi et al., 2012*); regulation of cryptic exon splicing may affect transcript levels in neurons and/ or prevent spurious expression of transcripts in non-neural cells. Another point of interest will be to assess whether there is cell-type specific regulation of PTC exons in specific neuronal types within the adult brain in a manner analogous to PTBP1 regulation of expression within different tissues of the body.

Here we show the frequent use of trans-acting control by a single neuronal splicing factor, NOVA, to regulate an array of target transcripts. This mechanism is biologically rich, as it includes both the up- and down-regulation of target RNAs through alternative splicing coupled to NMD in neuron. The regulatory roles of these actions are underscored by our observation that the cryptic NMD exons NOVA acts upon are evolutionarily conserved (Zhang and Darnell, manuscript in preparation), suggesting a regulatory rather than quality-control NOVA function. Taken together, our data suggest a model in which NOVA regulation maintains a balance of transcript levels through NOVA-dependent alternative splicing coupled NMD and, to a lesser degree, through direct interaction with 3′ UTR binding elements (*Figure 9*).

We found that NOVA mediated NMD-coupled splicing can be dynamically regulated in the brain. This suggests in turn that NOVA activity may be regulated by such dynamic events, leading to changes in RNA regulation and ultimately changes in neuronal proteins. The mechanisms that regulate NOVA are largely unknown, but interestingly, NOVA may be phosphorylated by GSK3 (*Dredge et al., 2005*), which itself is known to be inactivated in response to various extracellular stimuli in the brain, stimulating pathways such as the PI3K/Akt pathway (*Hur and Zhou, 2010*), which are activated by electrical activity (*Richter and Klann, 2009*). Thus it is possible that neuronal cell dynamics lead to fine tuning of RNA processing and expression through RNA binding proteins that act as sensors of electrical activity.

## NOVA-regulated NMD-coupled splicing and epilepsy

There is growing interest in the relationship between alternative splicing and neuronal excitability in the brain, as underscored by the recent finding that loss of the splicing factor Rbfox1 results in increased susceptibility to seizures and concomitant splicing changes in transcripts encoding synaptic proteins (*Gehman et al., 2011*). Interestingly, NOVA interacts functionally with Rbfox1 (*Zhang et al., 2010*), and many of the proteins showing large Nova-dependent changes are also involved in synaptic regulation with links to epilepsy were also found to have NOVA-dependent NMD exons regulated after induction of seizures in mice. For example, NOVA regulation of NMD splicing controls the levels of DLG3, one of a family of postsynaptic density proteins, including PSD95, that transduce signals from the NMDA receptor. Increases in both PSD95 (*Ying et al., 2004*) and DLG3 (*Qu et al., 2009*) in cortex or post-synaptic fractions, respectively, have been described in brains of patients with focal cortical dysplasia and epilepsy. NOVA regulation of NMD splicing also controls the levels of SCN9a, a sodium channel initially thought to act primarily in dorsal root ganglia (*Meisler et al., 2010*) (neurons that do not express NOVA proteins (*Buckanovich et al., 1993*)), but more recently found to be mutated in human epilepsy (causing familial febrile seizures; (*Singh et al., 2009*)). Other NOVA-NMD related targets in which gene mutations have been implicated in human epilepsy include the anion transporters SLC4a10 and SLC4a3 (*Table 1*; (*Gurnett et al., 2008*; *Sander et al., 2002*)).

The observation that either induction or loss of the same protein activity may be associated with epilepsy, suggest that the proper stoichiometry of synaptic proteins (or protein activity), rather than their absolute amount, may be a key determinant in electrical balance (*Figure 8*). This correlates with

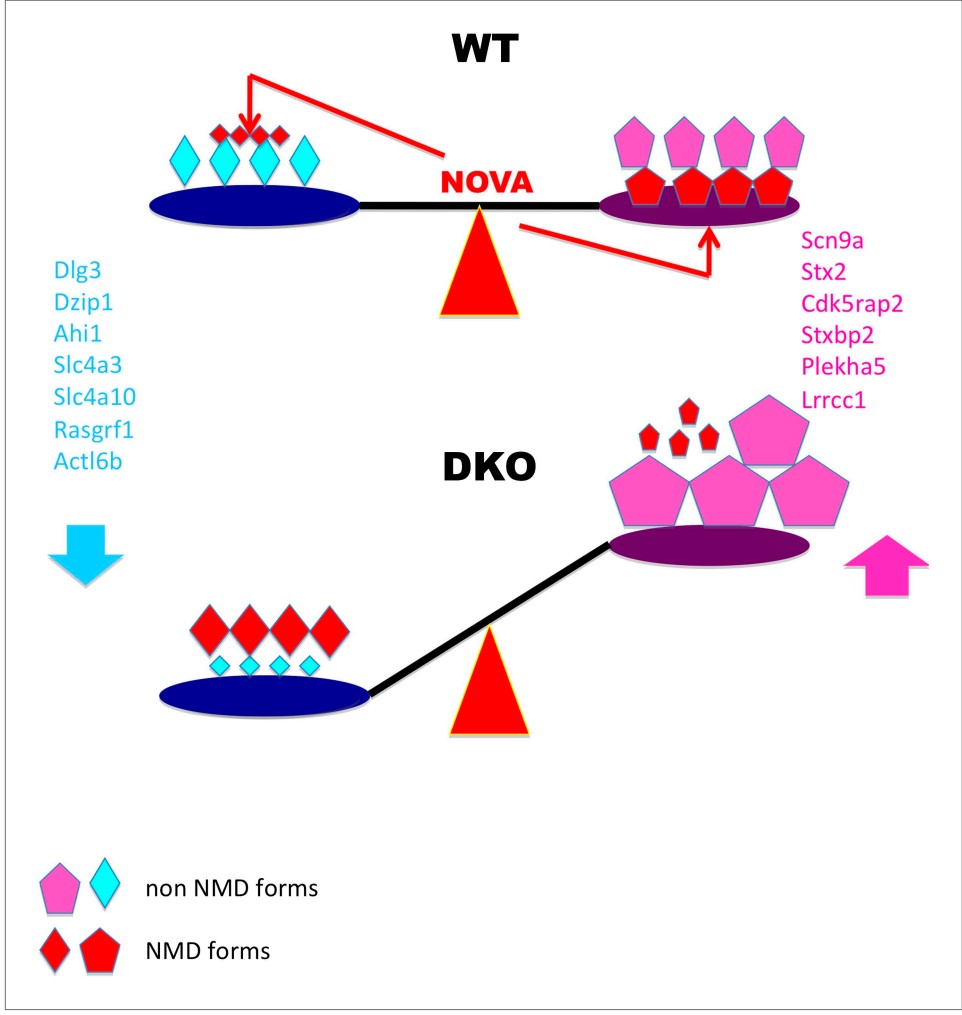

**Figure 9**. Model linking electrical activity with Nova-dependent splicing of cryptic NMD exons to maintain the balance of synaptic proteins. In WT brain, NOVA represses some cryptic NMD isoforms (small red diamonds on left) while promoting others (large red pentagons on right), thereby maintaining the balance of protein levels. To a lesser degree, NOVA also stabilizes transcripts through 3' UTR interactions (*Figure 8*). In DKO brain, the absence of NOVA disturbs this balance of protein in expression, contributing to aberrations in synaptic transmission. For example, Nova-regulation of cryptic NMD exons alters levels of the NMDA-receptor associated *Dlg3* (*Figures 2 and 3*) and sodium channel *Scn9a* (*Figure 4*) proteins, which are implicated in familial epileptic disorders and are dynamically regulated after seizures in mice (*Figure 6*).

our observation that NOVA haploinsufficiency is sufficient to induce spontaneous epilepsy (*Figure 6*), an observation also made with the neuronal RNABP Elavl3 and Elavl4 (*Ince-Dunn et al., 2012*). Moreover, NOVA control of NMD exons can act to either up or down-regulate protein levels generated from target transcripts, and, more specifically, to act after seizure to restore balance to synaptic proteins that would inhibit further seizure (e.g. increasing SCN9A, which would presumably be anti-epileptic). Hence the normal regulation of NOVA itself may provide an RNA-based means of regulating homeostatic plasticity in the face of electrical stress.

## New links between nuclear and cytoplasm RNA regulation

We have previously found that NOVA regulates neuronal alternative splicing, and that steady-state levels of these regulated transcripts are not significantly altered in *Nova* KO mice (*Ule et al., 2005b*; *Licatalosi et al., 2008*). The observation here of a significant set of transcripts undergoing NOVA-dependent regulation of steady-state levels on the surface therefore initially suggested that this was an independent set of NOVA actions. However, our finding that these were disproportionately

NOVA-mediated actions to maintain neuronal transcript levels (215 vs 14) is consistent with an overall action linking alternative splicing suppression of NMD exons to steady-state transcript levels.

Interestingly, our data also suggested a more general link between splicing and mRNA stability. Among transcripts with NOVA 3′ UTR binding we observed a significant number also had intronic NOVA binding, suggesting a possible hierarchy of events in which nuclear binding precedes (and potentially determines) 3′ UTR binding. Such connections were originally suggested from studies in *Drosophila*, in demonstration of the coupling of the splicing and localization of Oskar mRNA (*Hachet and Ephrussi, 2004*), and the observation that the predominantly nuclear Hrb27c protein (whose mammalian homologue is the DAZAP1 splicing factor; (*Goina et al., 2008*)) also binds to and is required for localization of *Grk* mRNA (*Goodrich et al., 2004*). In mammalian cells, ZBP1 binds to actin mRNA in the nucleus and subsequently is required for its localization (*Gu et al., 2002*; *Pan et al., 2007*), and NOVA binds to transcripts (e.g. GlyRα2) in introns to regulate alternative splicing and is also associated with 3′ UTR and localization of the same mRNA in the neuronal dendrite (*Racca et al., 2010*).

One potential explanation for such observations, as well as for the link between nuclear and cytoplasmic NOVA tags seen here, would be reassortment of transcripts in cis on individual transcripts, providing a means of linking nuclear functions such as alternative splicing with cytoplasmic function such as RNA localization or RNA stability. For example, intronic NOVA binding to high affinity intronic clusters would allow a high local concentration of the protein upon individual transcripts, potentially allowing rearrangement in cis and binding of additional secondary RNA elements, including those within the 3′ UTR. However, it should be noted that HITS-CLIP provides data that is a population average of many individual RNA–protein interactions, so that such potential cis-actions on a single transcript can only be inferred. Nonetheless, it is noteworthy that such local concentration and one-dimensional rearrangements *in cis* are believed to play an important role in scanning and initiation of transcription (*Blainey et al., 2006*; *Wang and Greene, 2011*), and hence may plausibly act on pre-mRNAs to distribute RNABPs *in cis* to coordinate different aspects of post-transcriptional regulation. In this context, it is especially notable that NOVA localization changes after seizure, suggesting that the association of NOVA with newly transcribed and spliced transcripts may be coupled in an activity-sensitive manner with their localization and/or translational control.

## Summary

We identified transcripts that show NOVA-dependent changes in steady-state levels in KO mice and used HITS-CLIP to drive the unexpected observation that what appeared to be functionless NOVA intronic binding sites in unregulated regions of these pre-mRNAs do function to regulate expression of cryptic NMD exons. These were enriched in transcripts encoding synaptic proteins, and their regulation mediated large changes in protein levels (up or down), including proteins implicated in familial epilepsy. The link between NOVA and regulation of excitation/inhibition balance was supported by finding that NOVA haploinsufficient mice had spontaneous epilepsy, and that pilocarpine-induced seizures regulated inclusion of a relatively small number of exons, a subset of which are regulated by Nova. Moreover, Nova redistributes from the nucleus to cytoplasm following pilocarpine seizure, and providing a possible mechanism underlying seizure-associated splicing changes involving Nova-regulated exons. Collectively, our data support a model in which changes in electrical activity affect the distribution of Nova, altering the splicing of cryptic exons that elicit NMD; these changes may provide a homeostatic means of regulating levels of transcripts and proteins associated with epilepsy.

## Materials and methods

### Plasmid constructs

YCAY elements were amplified by PCR and added into the pd1EGFP vector (Clontech, CA). Mutations to YCAY elements were made by using QuickChange site-directed mutagenesis kit (Stratagene, CA). Primers are listed in *Supplementary file 2*. RT-PCR products were cloned by using pGEM-T-easy Vector system (Promega, WI) and then sequenced (Genewiz, NJ).

### Nuclear/cytoplasmic fractionation

Two fractionation methods were used. In the first, P13 wild type mouse cortex was, dissociated and then UV irradiated for CLIP (3 × 400 mJ/cm²) in a Stratalinker (Stratagene), and fractionation was performed as recommended with a commercial kit (BioVision, CA). In the second method, P13 mice

were used. The brain cortex was dissected of white matter, chopped and irradiated (3 × 400 mJ/cm$^2$) in HBSS (Invitrogen, NY) containing 0.01 M HEPES (pH 7.3) in a Stratalinker. Tissue was Dounce homogenized in cold gradient buffer (20 mM HEPES [pH 7.4], 150 mM NaCl, 5 mM MgCl$_2$, 0.5 mM DTT, protease inhibitor [Roche, NJ] and RNasin Plus RNase inhibitor [Promega, WI]). The homogenized tissue was then treated with 0.5% NP-40 on ice and centrifuged 2× at 2000×$g$ for 10 min, and the resultant supernatant was collected as the cytoplasmic fraction.

## Western blot analysis

Protein extracts (nuclear/cytoplasmic fractions and WT vs DKO brain extracts from P13 and E18.5 whole brains, respectively) were resolved by 10% SDS-PAGE and transferred to PVDF membranes (Millipore, MA). Membrane was blocked and incubated with primary antibodies including goat anti-NOVA2; mouse anti-HSP90, mouse anti-HNRNP-C1, mouse anti-DLG3 (SAP102), mouse anti-SCN9a, rabbit anti-SYNGR3, rabbit anti-STX2 and mouse anti-γ-tubulin. Three different mouse brains were used as biological replicates as indicated throughout.

## RNA analysis

### Exon arrays in E18.5 mouse brain

For estimating the level of brain transcripts, total RNAs from four E18.5 mouse brains were extracted using Trizol (Invitrogen, NY) and RNAeasy kit (Qiagen, CA) and mRNA was amplified and labeled by the method provided by Affymetrix. Mouse MoEx 1.0ST Arrays were used for measuring signal intensity of each exon in the samples. To process the signals from the array, quantile normalization (WT vs DKO) and PM-GCBG (signal adjustment based on the background with similar GC content) were applied.

Steady state RNA levels were assessed using Affymetrix Mouse MoEx 1.0ST Exon arrays (Affymetrix, CA), and analyzed with Affymetrix Power Tools using IterPLIER. IterPLIER was used for selecting appropriate 'core exon' probes to estimate gene-level intensities. Finally, median log$_2$ values from four biological replicates are used to estimate level of transcripts in the E18.5 brain in WT vs DKO.

Interrogation and analysis of Affymetrix Exon junction arrays was undertaken as previously described, using the ASPIRE and ASPIRE2 programs (*Ule et al., 2005b*; *Ince-Dunn et al., 2012*).

### RT-PCR and qRT-PCR

cDNA was generated by using random hexamers and Superscript III (Invitrogen, NY). Once the number of PCR cycle was tested for the linear range, $^{32}$P-dCTP was added into the PCR reactions for the last two cycles and then run onto denaturing gels (6% polyarylamide/7 M urea). It was exposed to X-ray film (Kodak, NY). Quantitative RT-PCR (qRT-PCR) was performed by using a MyiQ single-color real time PCR detection system (Biorad, CA), using mouse tubulin as an internal control, and relative mRNA level (WT vs DKO) was calculated by ΔΔCt values. All qRT-PCR was performed in RNA from at least three biological replicates. qRT-PCR and other PCR primers are listed in *Supplementary file 2*.

### Northern blot analysis

Total RNA (10 μg) was electrophoresed and transferred into the Nylon membrane (Ambion, CA). The cDNA fragments complementary to *Dlg3* (digested by XmaI and BspE1), *Syngr3* (primers as below) and *Gapdh* (Ambion, CA) were labeled with $^{32}$P-dCTP using prime II random label kit (Stratagene, CA). The membrane was hybridized with $^{32}$P-labeled probes at 45°C overnight. After several washes, the membrane was exposed to X-ray film (Kodak, NY).

## HITS-CLIP and bioinformatics

NOVA HITS-CLIP was performed on subcellular fractions using previously described protocols (*Ule et al., 2003*, *2005a*; *Jensen and Darnell, 2008*; *Licatalosi et al., 2008*). High-throughput sequencing was performed on an Illumina GAIIx. CLIP tags were analyzed as previously described (*Licatalosi et al., 2008*), with modifications as described below.

### Genomic mapping of illumina reads

Sequencing reads were first filtered by the default Illumina pipeline. Filtered reads were then mapped by Eland using different sizes from 25 to 32 nt, with at most two mismatches (*Zhang et al., 2010*). A read is kept for analysis only if it was mapped to an unambiguous locus with at least one particular size. If unambiguous mapping was possible with different sizes, the one with largest size, yet without increasing number of mismatches was kept. Overall, 57% of reads are unambiguously mappable (*Table 2*).

For each individual CLIP experiments, we further collapsed reads with the same start genomic coordinates to remove potential RT-PCR artifacts. As a result, 1.3 million nuclear CLIP tags and 0.2 million cytoplasmic CLIP tags were unique and used for further analysis (*Table 2*).

## Definition of CLIP clusters

To generate a reliable definition of CLIP clusters, we used a cohort of ~20 independent NOVA CLIP experiments. CLIP clusters were identified by a Hidden Markov Model (HMM), which will be described in detail elsewhere. For each of the clusters, the number of CLIP tags for each nuclear and cytoplasmic CLIP experiment was counted. Only clusters with ≥2 tags in the nuclear/cytoplasmic tags were kept for this study (81,162 clusters). To measure the robustness of each cluster, biological complexity (BC) was defined as the number of samples with at least 1 tag in the cluster. BC was calculated for nuclear and cytoplasmic CLIP experiments, respectively. For each cluster, we also count the number of tags overlapping with each position and define peak height by the maximum number of overlapping tags.

GO analysis was performed by using DAVID (http://david.abcc.ncifcrf.gov/).

## Antibodies

Human sera from POMA patients was used to detect NOVA proteins in mixed cultured neurons. HSP-90 antibody (BD biosciences, CA), γ-tubulin (GTU88) antibody (Sigma, MO), hnRNP-C1/C2 antibody (Abcam, MA), NOVA2 antibody (Santa Cruz, CA) DLG3 (SAP102) and SCN9a antibodies (neuro-Mab, CA), STX2 antibody (Synaptic Systems, Germany), Neurofilament M antibody (Millipore, MA), UPF1 antibody (Dr Lykke-Andersen) and SYNGR3 antibody (Dr Janz) were used. All secondary antibodies mouse, rabbit, goat and human HRP- and Cy2/Cy3/Cy5-conjugated antibodies are from Jackson Immuno Research Laboratories, PA.

## Primary mouse neuronal culture and emetine treatment

Primary mouse neuronal cultures were generated from E18.5 cortical neurons prepared as previously described (*Eom et al., 2003*). In brief, forebrain from embryonic 18.5 days mouse brain (WT and DKO) was dissected, trypsinized, and plated on poly-l-lysine coated coverslips and dishes (BD biosciences, CA) in neurobasal medium (Invitrogen, NY) for immunofluorescence and cultured for 10 days at 37°C in 5% $CO_2$ and transfection with nucleofection was performed before plating. Emetine (100 μg/ml; Sigma, MO) was added to cells grown for 6 days in vitro (DIV) WT/DKO cultures for 10 hr.

Primary mouse neuronal cultures and N2a cells were transfected with siRNA ON-TARGETplus SMARTpool (Dharmacon, PA) and DY547 conjugated siRNAs by nucleofection (Lonza, NJ) as described by manufacturers. 48 hr after transfection with a SMARTpool of four siRNAs to *Upf1* or an equimolar amount of non-targeting control siRNAs (Dharmacon, PA), the cells were fixed in paraformaldehyde (4% in PBS) at room temperature for 15 min for immunofluorescence and RNA was extracted for RT-PCR.

## Immunofluorescence

Immunofluorescence was performed as described (*Racca et al., 2010*) with the following modifications. After fixation, coverslips were incubated for 1 hr at room temperature in Tri Buffered Saline (TBS) with BSA (2%), FBS (2%) and Triton X-100 (0.1%). Primary antibody incubations were for another 1 hr at the same condition with BSA (1%) and Triton X-100 (0.1%). All secondary antibodies were affinity-purified donkey antibodies to mouse, rabbit or human IgG conjugated to a Cy2/Cy3/Cy5 (Jackson Immuno Research Laboratories, PA). Coverslips were mounted with prolong gold antifade reagent (Invitrogen, NY) after DAPI (4,6-diamidino-2-phenylindole dihydrochloride) staining the cells.

## Pilocarpine treatment

Seizures were induced as follows: 8-week old mice were given subcutaneous injection of methyl scopolamine nitrate (1 mg/kg). After 30 min, mice were given subcutaneous injection of pilocarpine (320–340 mg/kg). After 2 hr, RNA was extracted from brain (pilo vs sham) and RT-PCR was performed.

## Chronic electroencephalographic (EEG) recordings

Adult *Nova2* (-/+) and wild-type (+/+) mice (aged 6–9 months) were implanted for chronic EEG recordings. Mice were anesthetized with Avertin (1.25% tribromoethanol/amyl alcohol solution, i.p.) using a dose of 0.02 ml/g. Teflon-coated silver wire electrodes (0.005 inch diameter) soldered to a microminiature connector were implanted bilaterally into the subdural space over frontal, temporal, parietal, and occipital cortices. Digital EEG activity was monitored daily for up to 2 weeks during prolonged

overnight and random 2 hr sample recordings (Stellate Systems, CA, Harmonie software version 6.0b). A digital video camera was used to simultaneously monitor behavior during the EEG recording periods. All recordings were carried out at least 24 hr after surgery on mice freely moving in the test cage.

## Additional information

### Competing interests
RBD: Reviewing Editor, *eLife*. The remaining authors have declared that no competing interests exist.

### Funding

| Funder | Grant reference number | Author |
|---|---|---|
| Howard Hughes Medical Institute | | Robert B Darnell |
| National Institutes of Health | NS34389 | Taesun Eom, Chaolin Zhang, Kenneth Lay, John Fak, Robert B Darnell |
| National Institutes of Health | HD24064 | Jeffrey L Noebels |
| National Institutes of Health | NS29709 | Jeffrey L Noebels |
| National Institutes of Health | NS081706 | Robert B Darnell |
| National Institutes of Health | NS40955 | Robert B Darnell, Huidong Wang |

The funders had no role in study design, data collection and interpretation, or the decision to submit the work for publication.

### Author contributions
TE, Acquisition of data, Analysis and interpretation of data, Drafting or revising the article; CZ, Analysis and interpretation of data, Drafting or revising the article; HW, Acquisition of data, Analysis and interpretation of data; KL, Acquisition of data; JF, Acquisition of data; JLN, Acquisition of data, Analysis and interpretation of data, Drafting or revising the article; RBD, Conception and design, Analysis and interpretation of data, Drafting or revising the article

### Ethics
Animal experimentation: This study was performed in strict accordance with the recommendations in the Guide for the Care and Use of Laboratory Animals of the National Institutes of Health. All of the animals were handled according to approved institutional animal care and use committee (IACUC) protocols of the Institutional Animal Care and Use Committee of Rockefeller University (protocol numbers 10064 and 10097).

## Additional files

### Supplementary files
- Supplementary file 1. List of validated targets that were alternatively spliced after pilocarpine induced status epilepticus.
- Supplementary file 2. PCR primers used in this work. All PCR primers are shown, oriented 5′ to 3′.

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
