## [Decision Letter]

eLife posts the editorial decision letter and author response on a selection of the published articles (subject to the approval of the authors). An edited version of the letter sent to the authors after peer review is shown, indicating the substantive concerns or comments; minor concerns are not usually shown. Reviewers have the opportunity to discuss the decision before the letter is sent (see review process). Similarly, the author response typicalsly shows only responses to the major concerns raised by the reviewers.

Thank you for choosing to send your work entitled “NOVA-dependent regulation of occult NMD exons controls synaptic protein levels after seizure” for consideration at *eLife*. Your article has been evaluated by a Senior Editor and 3 reviewers, one of whom is a member of *eLife's* Board of Reviewing Editors. The following individuals responsible for the peer review of your submission want to reveal their identity: Ben Blencowe and Doug Black.

The Reviewing Editor and the other reviewers discussed their comments before we reached this decision, and the Reviewing Editor has assembled the following comments based on the reviewers' reports.

In this manuscript, Eom and colleagues investigate the molecular basis by which the neural-specific splicing regulator Nova controls the levels of transcripts and corresponding proteins with important functions in the synapse. Using microarray profiling they detect ∼200 genes with increased transcript levels in the brains of wild-type mice compared to brains of *Nova1*/*Nova2* double knockout (DKO) mice. A much smaller number of genes show the opposite behavior. Using HITS-CLIP data, the authors examine the nuclear-cytoplasmic distribution of Nova binding sites in transcripts with differences in steady-state levels. While Nova binding to 3′UTR sequences appears to control the levels of some transcripts, a more widespread mechanism involves Nova binding within intronic sequences adjacent to previously unannotated exons, referred to as “occult exons” by the authors. Nova-dependent suppression or activation of the PTC-introducing exons promotes increased or decreased transcript/protein levels, respectively. The authors further use pilocarpine-induced seizures to investigate how changes in electrical activity might impact Nova-dependent regulation of transcript levels.

The results described are interesting and will appeal to researchers in the fields of gene regulation and neurobiology. Although regulation of transcript levels via alternative splicing coupled to NMD has already been described in several biological contexts, the identification of a coherent network of splicing events subject to this mode of regulation, with potentially important roles in the synapse, is a significant advance. Moreover, the use of HITS-CLIP to identify novel exons that function in regulating transcript levels is a clever application of an already powerful genome-wide approach. The reviewers have several concerns that they would like to see addressed in a revised manuscript.

1. The authors conclude from their experiments involving pilocarpine-induced seizures that the “clear increases in utilization of the NMD exon suggest that the levels of these transcripts can be dynamically tuned by electrical activity”. However, pilocarpine-induced seizures are known to cause oxidative damage and neuronal loss (e.g., Turski et al. 1986, which the authors cite). The authors should attempt to address whether this treatment directly impacts Nova-dependent splicing activity. At the very least, they should mention the caveats of this experiment, soften their conclusions, and modify the manuscript title.

2. A possible alternative model for the physiological relevance of the authors' findings is that the Nova-dependent suppression of cryptic exons serves as a mechanism to ensure that specific neuronal transcripts are not spuriously expressed in non-neuronal cells. The authors could readily test this by performing RT-PCR assays on the Nova-regulated transcripts, +/- *Upf1* knockdown, in non-neural cells. In this context, it is worth noting related work of Makeyev and co-workers (*G&D* 26, 1209; 2012), who have identified a class of Ptbp1-dependent retained introns that suppress expression of neural transcripts in non-neural cells.

3. The authors utilize a liberal cutoff in their identification of differentially expressed genes from their microarray analysis. A magnitude log_2_ change of 0.3 corresponds to slightly less than a 1.25 fold ratio change, and given the noise inherent in microarray measurements, it is unlikely that estimated changes in this range will validate at the same rate as those changes estimated to be >1.5 fold. Accordingly, the authors are requested to use qRT-PCR assays over the range of 229 genes predicted to show expression differences to assess false-positive rates. In this context, the descriptor “widespread” may be overstated.

---

## [Author Response]

*1. The authors conclude from their experiments involving pilocarpine-induced seizures that the “clear increases in utilization of the NMD exon suggest that the levels of these transcripts can be dynamically tuned by electrical activity”. However, pilocarpine-induced seizures are known to cause oxidative damage and neuronal loss (e.g., Turski et al. 1986, which the authors cite). The authors should attempt to address whether this treatment directly impacts Nova-dependent splicing activity. At the very least, they should mention the caveats of this experiment, soften their conclusions, and modify the manuscript title*.

We agree with the reviewers that being more circumspect here is prudent and we have revised the text to include caveats along the lines suggested. We specifically address the issue of whether neuronal loss could account for our data below and in the revised text (“Pilocarpine can induce neuronal loss days after seizure (Turski et al., 1986)…”).

We have taken the reviewers' advice and examined whether this treatment impacts Nova-dependent splicing activity more generally. We have considered carefully our response particularly in light of the editorial policy at *eLife*, which seems to be governed by good sense in providing consensus and focus to reviewer comments and efforts to avoid onerous burdens proposing new lines of research in response to the reviews. We applaud this approach, recognizing that unreasonable demands for additional manuscripts' worth of data may be made by other top-tier journals, and that *eLife* wishes to avoid such demands. At the same time, we interpret *eLife* policy as allowing a reasonable degree of discretion in the detail with which authors choose to respond to the reviewers' questions. In this case, we have responded to the reviewers' comment that we “should attempt to address” Nova's effects on splicing, which we interpret as a suggestion (and not a demand) by doing the full experiment, and we now report the results in this revision in tremendous detail. We include a large amount of very high quality experimental data that we were planning to use as the basis for a separate manuscript. Our intention here is to strengthen the paper significantly, and also to pay genuine respect the reviewers' concerns and to the ethos of the new *eLife* journal itself, by responding to the option to do additional work with the maximum, rather than minimum, response.

In the revised manuscript, we present new experiments analyzing splicing data in mouse brain in various regions (hippocampus and cortex), at different timepoints (4 and 24 hours) and using several platforms (exon junction arrays and RNA-Seq). We go on to experimentally validate this data, and to overlay this analysis with an examination of Nova-dependent splicing as requested. We supplement this validation of RNA levels by examining Nova-dependent cell biology with a completely new finding that may be impacting Nova's effects on the splicing of some transcripts. Finally, we document the physiologic relevance of the Nova-dependent/activity-dependent splicing and cell distribution changes in aggregate by showing that Nova2 KO and indeed, heterozygous mice, have spontaneous seizures.

These new data are presented in the sections entitled “Dynamic control of NOVA-regulated NMD exons”, “Cortical hyperexcitability and epilepsy in Nova2 heterozygous mice”, and “Activity-dependent changes in NOVA subcellular localization”. In sum, the new experiments address in molecular and cell biologic detail the effects of pilocarpine-induced seizures on Nova biology, and further tie Nova biology to naturally occurring seizures in the haploinsufficient state.

More precisely, in response to the reviewers' question, these new data allow us to conclude the following:

“We examined the 35 alternate exons that showed pilocarpine-dependent changes to see which were present on our list of Nova-dependent exons, and identified only 8 as Nova-regulated (Table 3). These data indicate that only a small fraction of potential regulated alternate exons, including those regulated by Nova, are affected by seizure activity, and that pilocarpine does not appear to have a global impact on Nova splicing activity.”

“Thus perturbing NOVA steady-state levels *in vivo* has relevant physiologic consequences, as haploinsufficiency is sufficient to give rise to spontaneous epilepsy in mice.”

“We consistently observed marked shifts of NOVA from the neuronal nucleus to the cytoplasm within 2-4 hours of pilocarpine-induced seizures … These data suggest that there may in part be a cell biologic mechanism whereby pilocarpine might affect the ability of NOVA to regulate splicing of some targets, but additional experimental follow-up of these observations will be required to fully understand their nature and consequences.”

These new data are presented in the following:

• Table 2: Global changes in splicing after pilocarpine seizure

• Table 3: Validation and splicing changes in Nova-regulated targets.

• Figure 6: Nova haploinsufficient mice have spontaneous epilepsy

• Figure 7: Pilocarpine seizures induce a change in Nova subcellular localization.

We note that the sorts of damage the reviewers are concerned about (e.g., Turski et al., as well as a number of subsequent papers) involve neuronal loss happening as a relatively distant event after the induction of pilocarpine seizures. In contrast, we are examining RNA by qRT-PCR 2 hours after seizure.

To be more explicit, Turski et al find evidence of neuronal damage at 5–7 days post pilocarpine seizure. The earliest they noted this was 48 hours. Moreover, they found that pre-treating animals with the muscarinic agonist scopolamine abrogated the severe neuronal damage; we pre-treated animals with 1mg/kg scopolamine 30 minutes prior to pilocarpine injection. Importantly, this pilocarpine protocol is generally used to induce kindling and long-term chronic epilepsy, which, as the reviewers point out, definitely induces long-term neuronal stress including death. On the other hand, the protocol also works as an acute stress—in fact, we used this protocol following discussion and recommendation from one of the world's experts in mouse epileptology, Jeff Noebels, who collaborated with us on a recent paper (Ince-Dunn et al., *Neuron*, 2012). Jeff is now included as a co-author on the current manuscript, along with Huidong Wang, given their role in generating and analyzing the more extensive data collection we are including here.

To more directly address the reviewers' concern, we have now performed immunofluorescence microscopy in the hippocampus, neocortex, and cerebellum, in a series of time points from 1 to 24 hours after pilocarpine-induced seizures (Figure 7). We find no evidence of neuronal loss. Interestingly, however, we noted a redistribution of Nova protein from the nucleus to the cytoplasm, an observation that is both cell-specific (evident in seizing cortical/hippocampus neurons, but not Purkinje neurons, which are not a focus of epileptic seizure) and protein-specific (we see no changes in Map2 staining).

*2. A possible alternative model for the physiological relevance of the authors' findings is that the Nova-dependent suppression of cryptic exons serves as a mechanism to ensure that specific neuronal transcripts are not spuriously expressed in non-neuronal cells. The authors could readily test this by performing RT-PCR assays on the Nova-regulated transcripts, +/- Upf1 knockdown, in non-neural cells. In this context, it is worth noting related work of Makeyev and co-workers (G&D 26, 1209; 2012), who have identified a class of Ptbp1-dependent retained introns that suppress expression of neural transcripts in non-neural cells*.

The reviewers make the interesting connection between the observations of Nova regulation and Ptbp1 regulation of transcript levels through regulation of PTC exons. The point is well made, specifically the idea that there could be cell-type specific regulation of PTC exons/protein levels within the nervous system in a manner analogous to Ptbp1 regulation of expression levels within different tissues of the body. This issue, including the Makeyev citation, has now been added to the text.

However, we do not believe that it is possible that Nova acts *in vivo* in the manner suggested, as there is no appreciable expression of Nova outside of neurons of the central nervous system (barring trace amounts of RNA detectable by RT-PCR only, whose significance is unclear, but may represent rare truncated transcript isoforms [unpublished data]).

In summary, while we could undertake transfections into non-neuronal cells, adding back exogenous Nova to demonstrate Nova-dependent and Upf1-dependent regulation of NMD exons, the experiment would not be as physiologically relevant as the experiments we have done. Specifically, we demonstrate in primary neurons expressing endogenous Nova, or in primary Nova DKO neurons, that the Nova-regulated splicing of NMD exons is required to control Upf1-mediated control of mRNA levels.

*3. The authors utilize a liberal cutoff in their identification of differentially expressed genes from their microarray analysis. A magnitude log2 change of 0.3 corresponds to slightly less than a 1.25 fold ratio change, and given the noise inherent in microarray measurements, it is unlikely that estimated changes in this range will validate at the same rate as those changes estimated to be >1.5 fold. Accordingly, the authors are requested to use qRT-PCR assays over the range of 229 genes predicted to show expression differences to assess false-positive rates. In this context, the descriptor “widespread” may be overstated*.

We appreciate the reviewers' point, and have now re-examined our validation data as requested. We present the new data in a revised Table 1. All qRT-PCR was done with 3 biologic replicates (3 animals) and three technical replicates (9 reactions per point). In these data we present analysis of 15 transcripts, with P-values and fold changes analyzed, that are examined using the original liberal cutoff, and, as requested, analysis of 7 new transcripts with much small changes, in the range requested by the reviewers. Perhaps because of the stringency with which we performed both the microarray analysis and the experimental validation, even here the small changes predicted by microarray were consistently seen with the qRT-PCR data, although the quantitative correlation was not as strong overall as for data showing larger changes. In any case, the basic data is now open for readers to review and judge for themselves; we would conclude that the false–positive rate, even for small changes, is relatively low.

To avoid any confusion, we deleted the descriptor “widespread”. We also rewrote relevant text, as in the following (which previously referred to “widespread”): “Taken together, these data indicate the extensive use of NOVA in steady-state neurons to inhibit cryptic NMD exons, and, to a lesser degree, mediate inclusion of NMD exons, as a means to maintain or reduce, respectively, the steady-state levels of neuronal transcripts and proteins.”